

# Construction of an Integrated Social Vulnerability Index in urban areas prone to flash flooding

Estefania Aroca-Jimenez[1], Jose Maria Bodoque[1], Juan Antonio Garcia[2], Andres Diez-Herrero[3]

[1]Department of Mining and Geological Engineering, University of Castilla-La Mancha, Avd. Carlos III, Toledo 45071, Spain
[2]Department of Business Administration, University of Castilla-La Mancha, Avd. Real Fabrica de Sedas, Talavera de la Reina 45600, Spain
[3]Geological Hazards Division, Geological Survey of Spain, 23 Calle Rio Rosas, Madrid 28003, Spain

*Correspondence to*: Estefania Aroca-Jimenez (estefaniacarmen.aroca@alu.uclm.es)

**Abstract.** Flash floods are considered to be one of the natural hazards with the greatest capacity to generate risk. Therefore, a change in traditional flood risk management (FRM) is necessary towards an integrated approach, which requires a comprehensive assessment of the social risk component. In this regard, integrated social vulnerability (ISV) gives us the spatial distribution, contribution and combined effect of exposure, sensitivity and resilience to the total vulnerability, although these components are often disregarded. ISV is characterized by the demographic and socioeconomic characteristics that condition a population's capacity to cope with, resist and recover from risk, and it can be derived from assessing the Integrated Social Vulnerability Index (ISVI). As far as we know, it has not yet provided a methodological approach to construct the ISVI in urban areas of Castilla y León (northern central Spain, 94,223 km$^2$, 2,478,376 inhabitants) prone to flash flooding. A hierarchical segmentation analysis (HSA) was performed prior to the principal components analysis (PCA), which helped to overcome sample size limitation inherent to PCA. ISVI was obtained from weighting vulnerability factors based on the tolerance statistic. Additionally, latent class cluster analysis (LCCA) was accomplished aiming to identify vulnerability spatial patterns within the study area. Our results show that the ISVI has high spatial variability. Moreover, the LCCA allowed us to identify the source of vulnerability in each urban area cluster. These findings enable a tailored design of FRM strategies, which intends to increase the efficiency of plans and policies, helping reduce implementation costs for mitigation measures.

## 1 Introduction

Flash floods are highly spatio-temporal localized flood events usually occurring in small, steep basins. They are caused by a sudden increase of the stream flow, generally due to spatially concentrated heavy rainfall and characterized by reaching a high peak flow in a short period of time (i.e., generally within six hours of rainfall) (Creutin et al., 2009; Gaume et al., 2009; Marchi et al., 2010; Liu et al., 2011; Borga, 2013; Wilhelmi and Morss, 2013; Bodoque et al., 2015; Terti et al., 2015). Its short duration, which limits or even voids any warning time, means that flash floods are considered to be one of the most



destructive hazards with the greatest capacity to generate risk, either in terms of socioeconomic impact or particularly concerning the number of casualties on a global scale (Marchi et al., 2010; Borga, 2013; Terti et al., 2015). In fact, according to Barredo (2007), 40% of flood-related casualties in Europe between 1950 and 2006 were caused by flash floods.

The growth of exposed population, the allocation of economical activities to flood-prone areas and the rise of extraordinary
event frequency over the last few decades (Huntington, 2006; Frigerio et al., 2016), have resulted in an increase of flash flood-related casualties and economic losses (Llasat et al., 2008; Marchi et al., 2010). The above highlight the need to make people aware of and prepared to live with risk (Birkmann, 2013). In this regard, the United Nations has invested a great deal of effort in promoting awareness on the importance of disaster reduction. The resulting initiative started in 1990 with the International Decade for Natural Disaster Reduction (IDNDR). The experience gained during this decade set foundations for
the International Strategy for Disaster Reduction, giving rise different frameworks, where the Sendai Framework for Disaster Risk Reduction 2015-2030 is the most topical (UNISDR, 2015). This new approach has enabled a change in FRM from an engineered-based perspective, that has proven not to respond effectively (Birkmann, 2013; Cutter et al., 2013; Koks et al., 2015) to a disaster-resilient perspective, highlighting the need for building resilient communities through integrating vulnerability reduction approaches into risk reduction policies (Cutter et al., 2008; Birkmann et al., 2013; UNISDR, 2015).

Many efforts were put in flood hazard analysis in past, but vulnerability assessment is still one of the biggest constraints in flood risk assessment to date (Mechler et al., 2014; Koks et al., 2015). A comprehensive vulnerability assessment should take into account any characteristics that increase susceptibility as well as any features that strengthen the population's ability to deal with extreme events (i.e., resilience) (Birkmann, 2013; Thieken et al., 2014). Numerous papers have analyzed the physical vulnerability component (Koks et al., 2014; Ocio et al., 2016). However, social aspects of vulnerability have
often been neglected (Cutter et al., 2003; Hummell et al., 2016), mainly due to the difficulty of quantifying variables that are inherently qualitative (Frazier et al., 2014). Social vulnerability tries to explain how a certain natural hazard produces an unequal impact on exposed population (Cutter et al., 2003; Nelson et al., 2015), and it can be characterized by any socioeconomic and demographic variables that influence society's preparedness, response and recovery (Birkmann, 2013; Cutter et al., 2013; Terti et al., 2015).

Social vulnerability has been assessed in different contexts (e.g., global environmental change, natural hazards). However, the number of works is reduced if we only focus on flood risk context (Tapsell et al., 2002; Burton and Cutter, 2008; Fekete, 2010; Mollah, 2016), and only very few studies are related to flash floods (Balteanu et al., 2015; Karagiorgos et al., 2016). Overall, social vulnerability analysis assesses vulnerability (Tapsell et al., 2002; Cutter et al., 2003; Nelson et al., 2015) and resilience separately (Cutter et al., 2008; Cutter et al., 2010; Siebeneck et al., 2015). Usually, the approach has been based on
calculating composite indices from sociodemographic and economic characteristics. Reductionist statistical techniques (i.e., generally factor analysis, FA, and principal components analysis, PCA) have been used for this purpose (Clark et al., 1998; Cutter et al., 2003; Dwyer et al., 2004; Fuessel, 2007; Grosso et al., 2015; Hummell et al., 2016; Rogelis et al., 2016).

Less attention has been paid to integrated analysis of vulnerability components, which considers the differential influence of exposure, sensitivity and resilience on total vulnerability (Frazier et al., 2014; Pandey and Bardsley, 2015; Weber et al.,



2015). The above helps to identify which characteristics contribute to vulnerability increases or decreases and where they are spatially represented. Accordingly, this approach provides a much more holistic assessment of vulnerability, since it models the combined effects of vulnerability components (Fuessel, 2007; Frazier et al., 2014). Moreover, it is worth mentioning that flash floods are produced in mountainous areas where data availability may be limited. Very few works take into account this constraint which, in some cases, is addressed by aggregating the variables being considered in order to obtain the vulnerability index (Balteanu et al., 2015). Determination of the vulnerability index requires vulnerability factor weighting. To this end, equal weights have usually been assigned to all factors (Cutter et al., 2003; Cutter et al., 2010), which may not lead to a realistic result (Frazier et al., 2014). Sometimes, differential weights according to expert judgments have been assumed (Zelenakova et al., 2015), which could be a limitation itself as experts' judgment may differ for the same issue (Asadzadeh et al., 2015). It is also worth mentioning statistical methods, such as correlation-based PCA (Mollah, 2016), which are being increasingly used (Asadzadeh et al., 2015).

This paper aims to calculate an integrated social vulnerability index (ISVI) to flash floods, which considers exposure, sensitivity and resilience. To address this objective, a set of variables were statistically analyzed firstly through hierarchical segmentation analysis (HSA) and secondly performing a PCA. The approach being implemented constitutes an alternative methodology to what is typically used in a social vulnerability assessment, enabling us to overcome the insufficient data availability frequently existing in many mountainous areas. Tolerance statistic was used as a variable weighting method. Additionally, latent class cluster analysis (LCCA) was performed in order to identify social vulnerability profiles within the study region.

## 2 Materials and Methods

### 2.1 Study area

The methodology proposed herein was implemented in the region of Castilla y León, which occupies almost all of central, northern Spain (Iberian Peninsula; Fig. 1). This region has a surface area of 94,230 km$^2$, making it the largest region not only in Spain but also in the European Union, exceeding the area of seventeen out of twenty-eight member states (e.g., Portugal, Austria or Belgium). Its relief is mainly composed of a large plateau between 700 and 1,100 m above sea level, surrounded by large mountain systems whose peaks can reach heights of up to 2,600 m. The climate is a continental variation of the Mediterranean type, with hot and dry Summers, but cold and relative dry Winters. Average annual rainfall ranges from 300 to 600 mm and primarily falls in Spring and Autumn, although more than 1,800 mm is not unusual in certain mountainous areas. High slopes, which limit the development of vegetation, and spatially concentrated heavy rainfall in certain mountainous areas, propitiate the triggering of flash floods. With regard to demographics, Castilla y León has a population of nearly 2.5 million, 5.5 % corresponding to foreign population. Population densities range from 9 to 65 inhabitants per km$^2$, giving a mean population density of 26 for the region, far lower than the mean for Spain, 92. The region is divided into 2,248 urban areas, with major differences between urban and rural urban areas. From the total number of urban areas, 94%



have less than 2,000 inhabitants, making up 26 % of the region's population. In addition, it is worth mentioning that the region demonstrates an ageing population, with an ageing index close to 2 (i.e., there are two people age 65 or older for each person below age 15). However, the ageing index is higher than 5 for urban areas with less than 2,000 inhabitants.

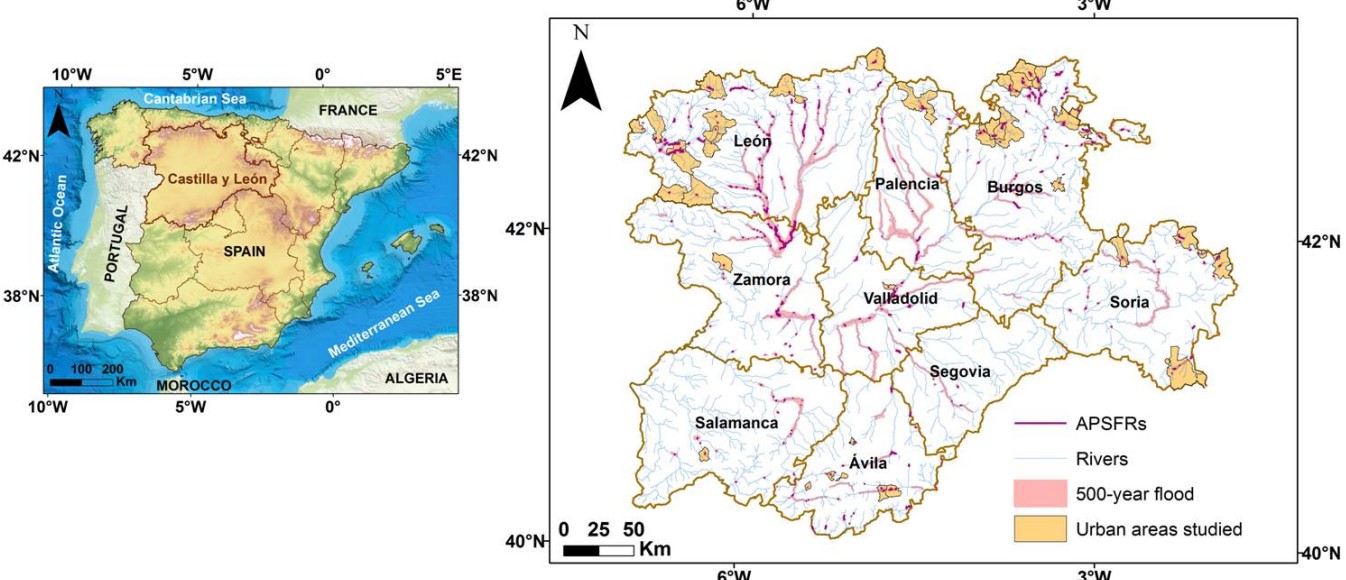

**Figure 1: Location of the region of Castilla y León.**

## 2.2 Methodological outline

First of all, those urban areas of the study region prone to flash flooding were distinguished. Next, a database of sociodemographic and socioeconomic variables was constructed for each of the urban areas considered (see Sect. 2.2.1.). It was not possible to start performing a PCA as it is usual in the literature to define the ISVI (Fekete, 2010; Frazier et al., 2014;Hummell et al., 2016), since the number of variables initially considered (i.e. 71 variables) outnumbered urban areas of interest (i.e. 39 urban areas) (Sarstedt and Mooi, 2014). Instead, a HSA was previously applied with the aim of dividing the database into small subsets of variables (see Sect. 2.2.2.). This allowed us to perform a PCA in each one of them enabling the extraction of the different vulnerability factors (see Sect. 2.2.3.), from which the ISVI can be composed (see Sect. 2.2.4.). In addition, vulnerability factors were used to identify social vulnerability patterns within the study area through a LCCA (see Sect. 2.2.5.) (Fig. 2).







**Figure 2: Methodological outline containing the different followed steps in the construction of the ISVI and the social vulnerability patterns.**

### 2.2.1 Identification of urban areas prone to flash flooding and database generation

Flash floods are produced in very specific areas, so it was necessary to specify the urban areas prone to this type of events. To do this, a number of simple but robust requirements were considered. The first requisite consisted of finding river reaches whose longitudinal slope crossing a given urban area was higher than 0.01 m m$^{-1}$ (Bodoque et al., 2015). To apply this criterion, a digital terrain model with a cell size of 200 m was used. It was provided by the Spanish National Geographic Institute (IGN) and was used as input data for the Geospatial Hydrologic Modelling extension (HEC-GeoHMS 10.0) (USACE, 2013), from which river longitudinal slopes were calculated. Secondly, and taking into account the river reaches selected in the previous step, we located the urban environments defined by Basin Water Authority as Areas with Potential Significant Flood Risk (APSFRs) (Caballero et al., 2011) and with low or exceptional probability (i.e., 500-year flood). They



were defined based on the preliminary flood risk assessment carried out by competent water, coasts and civil protection authorities, as stated by Directive 2007/60/CE on the assessment and management of flood risks. These areas were provided by the Spanish Ministry of Agriculture, Food and Environment (MAGRAMA). Subsequently, the aforementioned flooded areas were crossed with the river reaches selected according to the first criterion in order to identify the urban areas of interest, resulting in a total of 39.

Based on existing literature, a set of 71 variables was initially characterized for each of the 39 urban areas identified above. A total of 42 sociodemographic and socioeconomic variables were extracted from state, regional or local databases (e.g., population, education, buildings...). However, the other 29 variables were requested from certain public organisms or councils by means of telephone calls (e.g., dependency, development and infrastructures...) or generated through personal research (e.g., collective vulnerability, healthcare services...), particularly for smaller urban areas (i.e., with less than 150 inhabitants). These variables were then normalized to a percentage or per capita (Hummell et al., 2016). Redundant variables were removed after conducting a correlation test (Cutter et al., 2003). Specifically, those variables with a correlation coefficient above 0.9 were not considered. Therefore, 16 variables were eliminated from the database, continuing the methodological approach with the other 55 variables (Table 1). These were classified in 8 thematic information blocks: *i)* population (11 variables related to demography); *ii)* dependency (4 variables linked to elderly people); *iii)* education (2 variables associated with the level of educational attainment); *iv)* employment situation (5 variables related to unemployment status); *v)* healthcare services (8 variables linked to medical system characteristics); *vi)* development and infrastructures (10 variables associated with the economic potential of the region and its facilities); *vii)* buildings (13 variables related to construction features) and *viii)* collective vulnerability (2 variables linked to the availability of infrastructures to evacuate population).





**Table 1: Set of variables used in the exploratory analysis of social vulnerability dimensions.**

| Category | Variable | Description | Data source |
|---|---|---|---|
| Population | TPOP | Total population | Spanish Statistics Institute (2014) |
| | FOREIG | Foreigners | Spanish Statistics Institute (2014) |
| | POP0 | Inhabitants aged 0-4 | Spanish Statistics Institute (2014) |
| | POP5 | Inhabitants aged 5-14 | Spanish Statistics Institute (2014) |
| | POP15 | Inhabitants aged 15-64 | Spanish Statistics Institute (2014) |
| | POP65 | Inhabitants aged 65 or older | Spanish Statistics Institute (2014) |
| | PROJ_0 | Population projection aged 0-4 for 2025 | Spanish Statistics Institute (2014) |
| | PROJ_5 | Population projection aged 5-14 for 2025 | Spanish Statistics Institute (2014) |
| | PROJ_15 | Population projection aged 15-64 for 2025 | Spanish Statistics Institute (2014) |
| | PROJ_65 | Population projection aged 65 or older for 2025 | Spanish Statistics Institute (2014) |
| | NRESID | New residents | Spanish Statistics Institute (2014) |
| Dependency | DISABLED | Disabled people | Institute of Social Services and the Elderly (2013) |
| | DEPRAT_M | Dependency rates: males | Spanish Statistics Institute (2014) |
| | DEPRAT_F | Dependency rates: females | Spanish Statistics Institute (2014) |
| | HOUSE_OLD | Households where people aged 65 or older live | Population and Housing Census (2011) |
| Education | ILLITER | Illiterate people | Population and Housing Census (2011) |
| | LITER | Literate people | Population and Housing Census (2011) |
| Employment situation | LT_UNEMP | Long-term unemployed people | Regional Employment Observatory (2015) |
| | UN_RAT | Unemployment rates | Spanish Public Employment Service (2014) |
| | HS_0EMP | Households where any employed people live | Population and Housing Census (2011) |
| | HS_0UNEMP | Households where any unemployed people live | Population and Housing Census (2011) |
| | WORK_M | People that work within urban area of residence | Population and Housing Census (2011) |
| Healthcare services | HEALTH_C | Health centres | Spanish Ministry of Health, Social Services and Equality (2015) |
| | DIST_HC | Distance to the nearest health centre | Personal research (2015) |
| | TIME_HC | Travel time to the nearest health centre | Personal research (2015) |
| | TYPE_H | Type of healthcare (continuity) | Personal research (2015) |
| | H_BEDS | Hospital beds | Regional Statistics Information System (2014) |
| | DIST_H | Distance to the nearest hospital | Personal research (2015) |
| | TIME_H | Travel time to the nearest hospital | Personal research (2015) |
| | MED_ST | Medical staff | Personal research (based on council information, 2015) |
| Development and infrastructures | KINDERG | Kindergartens | Spanish Ministry of Education, Culture and Sport (2015) |
| | ELEM_SCH | Elementary schools | Spanish Ministry of Education, Culture and Sport (2015) |
| | SEC_SCH | Secondary schools | Spanish Ministry of Education, Culture and Sport (2015) |
| | RET_HOME | Retirement homes | Spanish Ministry of Health, Social Services and Equality (2009) |
| | TOUR_AC | Tourist accommodation | Regional Statistics Information System (2014) |
| | CAMPSITES | Campsites | Personal research (2015) |
| | DEBTS | Municipal debt per inhabitant | Spanish Ministry of Finance and Public Administrations (2014) |
| | PC_INC | Per capita income | Spanish Institute for Fiscal Studies (2011) |
| | FIX_INV | Fixed investments per inhabitant | Spanish Ministry of Finance and Public Administrations (2014) |
| | BUDGET | Municipal available budget per capita | Spanish Ministry of Finance and Public Administrations (2014) |
| Buildings | BUILTAREA | Built-up area per area without buildings | Spanish cadastre (2015) |
| | ABOVE_GR | Above ground built-up area | Spanish cadastre (2015) |
| | UNDER_GR | Underground built-up area | Spanish cadastre (2015) |
| | POP_SETL | Population per settlement area | Spanish cadastre (2015) |
| | CON_AGE | Mean age of household construction | Spanish cadastre (2015) |
| | PERM_H | Permanent households | Population and Housing Census (2011) |
| | VACANT_H | Vacant households | Population and Housing Census (2011) |
| | NONACCES | Non-accessible households | Population and Housing Census (2011) |
| | POORCOND | Households in poor condition | Population and Housing Census (2011) |
| | GDCOND | Households in good condition | Population and Housing Census (2011) |
| | ST_AGBG | Households with 1 storey above ground level and/or another storey below ground level | Population and Housing Census (2011) |
| | ST_AGL | Households with 2 or more storeys above ground level | Population and Housing Census (2011) |
| | USE_AR | Households' mean useful area | Population and Housing Census (2011) |



| Collective vulnerability | INTERS EVACUAT | Potential intersections between evacuation routes and rivers | Personal research (2015) |
|---|---|---|---|
| | | Areas suited to population evacuation | Personal research (2015) |

### 2.2.2 Exploring the dimensions of social vulnerability

Variables being considered were grouped together using the HSA application, using SPSS (IBM-SPSS v.19) statistical software. It is a multivariate statistical technique for automatic data classification attempting to divide an initial set of variables into different groups. This division is based on minimizing the distance among variables in the same group and maximizing the distance among variables in different groups (Sarstedt and Mooi, 2014). The division of variables into groups follows a hierarchical process. Initially, as many groups as variables were considered. Subsequently, successive iterations of hierarchical algorithms enabled variables to join up in larger groups. Once the variables were standardized (Cutter et al., 2003), the squared Euclidean distance was used as a distance measure, i.e., the square of the square root of the sum of the differences between variable values. In addition, Ward's method was used as a grouping method. It seeks the least possible variability within each group (i.e., the minimum variance), as an associative hierarchical algorithm which has been demonstrated to be one of the most effective (Pérez, 2004), especially when the sample size is small (Martín et al., 2015). In order to determine the number of groups, not only was the distance at which groups were differentiated into the graphical output of the HSA (i.e., the dendrogram) taken into account but also that variables contained in them were consistent and homogeneous in number. Finally, distinguishing groups of variables allowed us to conduct a principal components analysis in each of them.

### 2.2.3 Identification of vulnerability factors

SPSS (IBM-SPSS v.19) was used to implement Principal Component Analysis (PCA) in each group differentiated by the HSA. It aimed to reduce the number of variables into latent variables which are not directly observable, so-called principal components or factors, which are a linear combination of primitive variables (Sarstedt and Mooi, 2014). The Kaiser-Meyer-Olkin (KMO) statistic and Barlett's test of sphericity were estimated in order to evaluate the suitability of performing PCA in variables being considered. For each group, all the variables were initially considered in the factor extraction process. Variables with a low communality (values below 0.5) were subsequently removed and the factor extraction process was repeated until all variables had communality values above 0.5. Communality indicates how much variance of each variable can be reproduced by means of factor extraction. In cases in which a group set out more than one factor, they were separated and a PCA was performed with each one of them individually. Factor loadings represent the correlations between the factors and variables, which helped us to name each factor. Finally, factor scores were produced using the regression method, which is the most used (Sarstedt and Mooi, 2014). Factor scores embody a linear combination of the primary variables. Thus, each urban area was composed of as many factor scores as social vulnerability factors identified.



### 2.2.4 Construction of the Integrated Social Vulnerability Index (ISVI)

Factor scores for each vulnerability factor were saved as new attributes in the data file, which allowed us to use them for index construction. Factor scores were standardized and taking positive or negative values depending on whether an urban area exhibits the characteristic described by a certain factor above or below average (Sarstedt and Mooi, 2014).

Traditionally, factors that express sensitivity or exposure are considered as positive values in the ISVI; while factors that state resilience are considered as negative values (Cutter et al., 2013; Frazier et al., 2014; Hummell et al., 2016). In order to maintain this criterion, the sign of some factor scores were reversed (i.e., multiplied by -1).

ISVI for each urban area was calculated according to the following Eq. (1) (modified from Frazier et al. (2014)):

$$ISVI = E + S - R \,, \tag{1}$$

where ISVI is the integrated social vulnerability index, E is the exposure, S is the sensitivity, and R is the resilience. Each vulnerability component was estimated using Eq. (2) (modified from Frazier et al. (2014)):

$$V_C = \sum_{f=1}^{n} w_f \cdot S_f \,, \tag{2}$$

where $V_C$ is the vulnerability component (exposure, sensitivity or resilience), $w_f$ is the weight allocated to n factor, and $S_f$ are the factor scores of n factor.

The value of a certain vulnerability component was the sum of all factor scores of those factors within it multiplied by their respective weights. The weighting method used was based on the tolerance statistic. Tolerance is a statistical test to detect multicollinearity (Sarstedt and Mooi, 2014). Tolerance reaches a maximum value of 1 when a factor has no degree of multicollinearity with the other factors and a minimum value of 0 when a factor is a perfect linear combination of the others. Thus, vulnerability factors that expressed less redundant information would have more weight in the ISVI.

### 2.2.5 Identification of social vulnerability patterns

LCCA is a model-based clustering approach that was implemented (using Latent Gold® 4.5) with the purpose of identifying social vulnerability patterns within the study area. Urban areas of interest were classified into clusters (Vermunt and Magidson, 2002), where the statistical model showed the sources of vulnerability for each cluster. This classification was made by creating a latent categorical variable which measures the probability of belonging to a certain cluster according to

the characteristics of the vulnerability factors. Factor scores were used as indicators in order to identify the different clusters. A z-standardization of factor scores was implemented before introducing them into the statistical software. Five models integrating from one to five clusters were considered. Information criteria based on the model log likelihood BIC (i.e., Bayesian Information Criterion) and CAIC (i.e., Akaike's Information Criterion) were used as model selection tools in order to choose the best model, based on the minimum values of these two criteria (Morin et al., 2011).



## 3 Results

### 3.1 Integrated Social Vulnerability Index (ISVI)

The dendrogram shows the five groups of variables that were differentiated by the HSA (Fig. 3). Groups were homogeneous both in number of variables (each comprising between 10 and 13 variables) and in type of variables included. The first group

5 contained variables mainly related to large collective buildings. The second group of variables was connected to the kind of constructions and the economic potential of the region. The third group was related to demographic characteristics and the employment situation in the region. The fourth group of variables was primarily associated with elderly population. Finally, the fifth group did not show a clear dominance of any variables over others, although certain variables displayed significant correlation (i.e., $p < 0.05$).

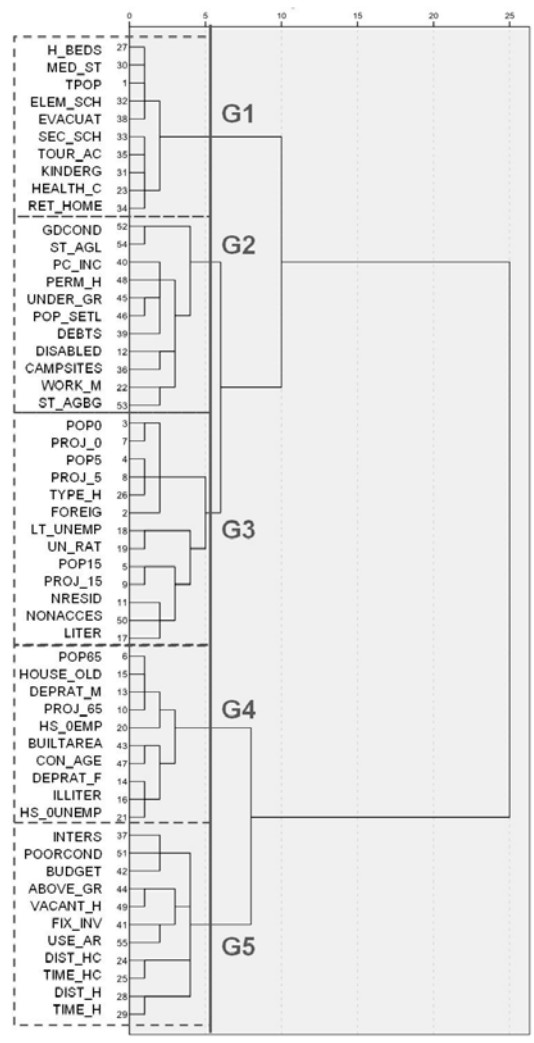

**Figure 3: Dendrogram resulting from the HSA. Each rectangle corresponds to an identified group, with a total of five groups (G1, G2, G3, G4 and G5). The meaning of the variables contained in each group is included in Table 1.**



Once variables with a low communality were removed (i.e. 17 variables), a PCA was performed in each group of variables, which enabled to identify 11 vulnerability factors  containing a total of 38 variables (Table 2): *1*) total social exposure; *2*) exposure in the urban built-up environment; *3*) constructive resilience; *4*) constructive exposure; *5*) youth social sensitivity; *6*) mature social resilience; *7*) labour social sensitivity; *8*) social sensitivity due to dependency; *9*) economic resilience due to investments; *10*) social hospital sensitivity; and *11*) social health sensitivity. In all cases, the KMO scores were higher than 0.5 and the Barlett's test of sphericity values were significant (i.e., p < 0.05). For vulnerability factors comprising two variables, the value of the correlation coefficient is indicated instead of the KMO score. Correlation coefficients for these vulnerability factors are considered significant (i.e., p < 0.05). In addition, all identified vulnerability factors show a percentage of explained variance over 70%. Factor loadings were used to allocate names to the vulnerability factors. This allowed us to classify vulnerability factors as they express exposure, sensitivity or resilience. Consequently, factors number one, two and four were considered to state exposure; factors number five, seven, eight, ten and eleven express sensitivity; and factors number three, six and nine were considered to state resilience.



**Table 2: Vulnerability factors identified with the PCA and additional statistical information (PCA results). The sign of the variable loadings indicates whether the correlation among variables that make up a certain vulnerability factor is positive or negative.**

| Factor | Variables | | KMO[a] / Correlation coefficient[b] | Explained variance | Loadings | Factor Name |
|---|---|---|---|---|---|---|
| 1 | TPOP | Total population | | | 0.968 | |
| | HEALTH_C | Health centres | | | 0.886 | |
| | H_BEDS | Hospital beds | | | 0.944 | |
| | MED_ST | Medical staff | | | 0.949 | |
| | KINDERG | Kindergartens | 0.846[a] | 83.68 | 0.918 | Total Social Exposure |
| | ELEM_SCH | Elementary schools | | | 0.974 | |
| | SEC_SCH | Secondary schools | | | 0.932 | |
| | RET_HOME | Retirement homes | | | 0.757 | |
| | TOUR_AC | Tourist accommodation | | | 0.886 | |
| 2 | POP_SETL | Population per settlement area | | | 0.777 | |
| | VACANT_H | Vacant households | 0.681[a] | 70.82 | -0.932 | Exposure in the Urban Built-Up Environment |
| | UNDER_GR | Underground built-up area | | | 0.788 | |
| | PERM_H | Permanent households | | | 0.860 | |
| 3 | GDCOND | Households in good condition | -0.431[b] | 71.54 | 0.846 | Constructive Resilience |
| | BADCOND | Households in poor condition | | | -0.846 | |
| 4 | ST_AGBG | Households with 1 storey above ground level and/or another storey below ground level | -0.655[b] | 82.73 | 0.910 | Constructive Exposure |
| | ST_AGL | Households with 2 or more storeys above ground level | | | -0.910 | |
| 5 | POP0 | Inhabitants aged 0-4 | | | 0.898 | |
| | POP5 | Inhabitants aged 5-14 | 0.720[a] | 78.03 | 0.901 | Youth Social Sensitivity |
| | PROJ_0 | Population projection aged 0-4 for 2025 | | | 0.827 | |
| | PROJ_5 | Population projection aged 5-14 for 2025 | | | 0.906 | |
| 6 | POP15 | Inhabitants aged 15-64 | 0.791[b] | 89.55 | 0.946 | Mature Social Resilience |
| | PROJ_15 | Population projection aged 15-64 for 2025 | | | 0.946 | |
| 7 | UN_RAT | Unemployment rates | | | 0.912 | |
| | LT_UNEMP | Long-term unemployed people | 0.692[a] | 75.95 | 0.872 | Labour Social Sensitivity |
| | HS_0UNEMP | Households where any unemployed people live | | | -0.828 | |
| 8 | POP65 | Inhabitants aged 65 or older | | | 0.944 | |
| | PROJ_65 | Population projection aged 65 or older for 2025 | | | 0.828 | |
| | DEPRAT_M | Dependency rates: males | 0.704[a] | 70.01 | 0.819 | Social Sensitivity due to Dependency |
| | DEPRAT_F | Dependency rates: females | | | 0.860 | |
| | HOUSE_OLD | Households where people aged 65 or older live | | | 0.808 | |
| | ILLITER | Illiterate people | | | 0.748 | |
| 9 | FIX_INV | Fixed investments per inhabitant | 0.782[b] | 89.08 | 0.944 | Economic Resilience due to Investments |
| | BUDGET | Municipal available budget per capita | | | 0.944 | |
| 10 | DIST_H | Distance to the nearest hospital | 0.863[b] | 93.16 | 0.965 | Social Hospital Sensitivity |
| | TIME_H | Travel time to the nearest hospital | | | 0.965 | |
| 11 | DIST_HC | Distance to the nearest health centre | 0.869[b] | 93.43 | 0.967 | Social Health Sensitivity |
| | TIME_HC | Travel time to the nearest health centre | | | 0.967 | |

Factor scores for each vulnerability factor were depicted using the quintiles classification (i.e., 20, 40, 60 and 80 percentiles)
with 5 classes: *i)* very low; *ii)* low; *iii)* medium; *iv)* high and *v)* very high (Fig. 4). For exposure and sensitivity factors, very
high categories correspond to red colours while for resilience factors, very high categories correspond to blue colours.



**Figure 4: Factor scores for identified vulnerability factors.**

Figure 5 illustrates, on the one hand, the ISVI value for each urban area using the quintiles classification. In this regard, ISVI

has high spatial variability, defining values that range from 0.085 to -0.055. Urban areas with highest ISVI values are mainly

concentrated in the northwest, while urban areas with the lowest values are found in the east and northeast of Castilla y

León. Each urban area has an associated bar chart showing the decomposition of each ISVI value into its components



(exposure, sensitivity and resilience). The direction of the bar indicates whether the sign of the vulnerability component is negative or positive. The height of the bar depicts the value of the vulnerability component (i.e., each vulnerability component was calculated by combining any vulnerability factors which contributed to each component, considering factor scores and the different weights). In addition, the numbers located in each bar show the categories based on the classification of quintiles in which each vulnerability component is found. Number 1 is associated with a very low category (i.e., very low exposure, sensitivity and resilience) and number 5 with a very high category (i.e., very high exposure, sensitivity and resilience).

**Figure 5: ISVI values and its decomposition into vulnerability components.**



## 3.2 Social vulnerability patterns

After performing the LCCA considering from one (sample homogeneity) to five clusters (sample heterogeneity with 5 patterns), the minimum values of BIC and CAIC determined that the optimum number of clusters of urban areas was three (Table 3).

**Table 3: Model fit summary of latent class cluster models initially considered.**

|  | Log-likelihood (LL) | BIC (LL) | CAIC (LL) | Number of parameters |
|---|---|---|---|---|
| One cluster | -603.153 | 1286.904 | 1308.904 | 22 |
| Two clusters | -501.287 | 1167.435 | 1212.435 | 45 |
| **Three clusters*** | **-440.610** | **1130.342** | **1198.342** | **68** |
| Four clusters | -407.146 | 1147.675 | 1238.675 | 91 |
| Five clusters | -381.371 | 1180.388 | 1294.388 | 114 |

**\*Best model according to BIC and CAIC**

In order to evaluate the vulnerability factor usefulness in terms of identifying these patterns, the parameters for each identified cluster are shown in Table 4. Neither the "social hospital sensitivity" factor nor the "economic resilience due to investments" factor was statistically significant when discriminating among the three clusters of urban areas ($p > 0.05$, highlighted in bold in Table 4). The percentage given under each cluster title shows the proportion of urban areas making up each cluster.

**Table 4: Parameters of urban area clusters associated with vulnerability factors. Vulnerability factors are sorted by vulnerability component (exposure, sensitivity and resilience)**

| Vulnerability factors | Cluster 1 (51.1%) | Cluster 2 (30.9%) | Cluster 3 (18.0%) | Robust Wald statistic | p-value | R² |
|---|---|---|---|---|---|---|
| Factor 1: Total Social Exposure | -0.295 | -0.549 | 0.844 | 9.794 | 0.008 | 0.244 |
| Factor 2: Exposure in the Urban Built-Up Environment | -0.054 | -0.708 | 0.779 | 37.071 | 0.000 | 0.428 |
| Factor 4: Constructive Exposure | 0.659 | -0.253 | -0.406 | 13.871 | 0.001 | 0.240 |
| Factor 5: Youth Social Sensitivity | -0.012 | -0.998 | 1.010 | 30.913 | 0.000 | 0.478 |
| Factor 7: Labour Social Sensitivity | 0.487 | -0.740 | 0.253 | 16.665 | 0.000 | 0.303 |
| Factor 8: Social Sensitivity due to Dependency | 0.092 | 0.706 | -0.798 | 38.442 | 0.000 | 0.262 |
| **Factor 10: Social Hospital Sensitivity** | **0.309** | **0.302** | **-0.611** | **3.363** | **0.190** | **0.127** |
| Factor 11: Social Health Sensitivity | -0.012 | 0.860 | -0.848 | 34.559 | 0.000 | 0.350 |
| Factor 3: Constructive Resilience | -0.020 | -0.418 | 0.438 | 6.017 | 0.049 | 0.087 |
| Factor 6: Mature Social Resilience | -0.204 | -0.501 | 0.704 | 20.442 | 0.000 | 0.174 |
| **Factor 9: Economic Resilience due to Investments** | **-0.307** | **0.593** | **-0.286** | **3.740** | **0.150** | **0.175** |



Finally, Figure 6 shows the three different clusters of identified urban areas, which help us to characterize the profile of each detected pattern. Moreover, each cluster is associated with a bar chart which depicts the cluster profile and it is depicted over the most representative urban area of each of them, calculated by taking into account the number of coincidences among the signs and the minimum distances between factor scores for each urban area and the mean factor scores for each identified

cluster. These bar charts gather the standard deviation values from the mean for each vulnerability factor, values that are located above each bar. The direction of the bar is related to the sign of these standard deviation values. That is, for factors that express exposure or sensitivity, positive values mean more exposure or sensitivity than the mean of the cluster. However, for factors that express resilience, positive values mean more resilience than the mean of the cluster. Thus, each cluster can be characterized:

Cluster 1 comprises 51.1% of urban areas of interest (i.e., a total of 20) and it is characterized by being made up of urban areas with the highest levels in the constructive exposure and the labour social sensitivity factors. This means that households usually have potentially flood-prone storeys (i.e., one storey above ground level and/or another storey(s) below ground level) and that urban areas have a bad employment situation (i.e., high rates of unemployment).

Cluster 2 comprises 30.9% of urban areas (i.e., a total of 12). It contains urban areas with the highest levels of social health

sensitivity and of social sensitivity due to dependency factors. This means that inhabitants that live there are usually elderly people and they have no health centres close by. On the other hand, these urban areas present the lowest levels in the youth social sensitivity factor, meaning that young population is scarce in these areas; and in the labour social sensitivity factor, which means that they have low unemployment rates. Moreover, urban areas with the lowest levels in the total social exposure and the exposure in the urban built-up environment are included here. This means that they are small urban areas

with no great urban development and where there is usually no presence of collective buildings such as hospitals or tourist accommodation. Regarding resilience factors, cluster 2 contains urban areas with the lowest levels in the constructive resilience and mature social resilience factors. This is due to the poor condition of households (e.g., cracks or damp) and the lack of middle-aged population (i.e., ageing populations).

Cluster 3 comprises 18.0% of urban areas (i.e., a total of 7). It is characterized by being made up of urban areas with the

highest values in the total social exposure, exposure in the urban built-up environment and youth social sensitivity factors. This means that they are populated areas with a large number of collective facilities (e.g., health centres or schools, among others), with great urban development and where population below age 14 (i.e., young population) is significant. In spite of the above, it is worth mentioning that these urban areas present the highest values in the constructive resilience factor and the lowest values in the constructive exposure factor, which means that constructions tend to be in good condition and they have

the highest proportion of households with two or more storeys above ground level, both characteristics that could help to reduce flood losses. On the other hand, these urban areas present the lowest values in the social sensitivity due to dependency and social health sensitivity factors, meaning that population is not ageing and urban areas have health centres within the municipal district. Moreover, these urban areas present the highest values of mature social resilience factor which means that the middle-aged population proportion is large and they could provide assistance during an extreme event.



It seems that there is a relationship between the ISVI and the clusters to which urban areas belong to Figure 6. In this regard, there are only significant differences between the ISVI values of clusters 1 and 2 (i.e., $p < 0.05$; ANOVA analysis). Moreover, it is verified that cluster 1 urban areas are more vulnerable than cluster 2 urban areas, with an ISVI mean value of 0.013 and -0.017, respectively.

**Figure 6: Characteristics of the urban areas that form the identified clusters. Bars with a meshed plot represent factors which were not statistically significant in the discrimination of clusters of urban areas. Bars are sorted by vulnerability component (exposure, sensitivity and resilience).**

## 4 Discussion

### 4.1 Data sources and methodology

Flash floods usually affect small mountainous urban areas (Marchi et al., 2010; Terti et al., 2015). Generally, the information available in these areas is limited, either because it is not available in public databases (i.e., it should be requested from the different councils) or because it is not generated at this work scale (i.e., it should be estimated from a bigger work scale),



which means a limitation in accomplishing any assessment related to flash floods (Ruin et al., 2009). However, this constraint is not usually presented for studies to characterize fluvial floods since they frequently affect significant urban areas in terms of population, usually meaning that more data and a larger number of event records are available.

This lack of information could condition the selected work scale which should coincide with the flood risk mitigation planning scale (Cash and Moser, 2000). An insufficient work scale could lead to the implementation of homogeneous vulnerability reduction measures in areas where vulnerability spatial variability is high which would reduce their effectiveness or might not provide a uniform vulnerability reduction (Eakin and Luers, 2006; Frazier et al., 2014). Here, the selected work scale was the urban area, as this entity in the region of Castilla y León tends to be small and homogeneous.

Furthermore, sensitivity and resilience are usually considered as static components (i.e., results capture a snapshot of vulnerability), when in fact they vary over space and time (Cutter et al., 2003; Eakin and Luers, 2006). The identification of spatial patterns here represents a step forward in that direction, which means an improvement of the FRM at regional scale. However, regarding temporal variability, we suggest periodic monitoring of variables identified as an explanation to social vulnerability to flash floods. This would allow urban areas to know how SVI values behave over time by means of periodic recalculation.

Concerning the ISVI calculation, it is critical to consider that ISVI values are not absolute. This means that the ISVI can be used to qualitatively compare whether an urban area is more vulnerable than others or in what proportion (Cutter et al., 2013). As regards the methodology proposed here, the preliminary implementation of a HSA helps to overcome the PCA sample size limitation (Sarstedt and Mooi, 2014). Regarding the above, most published works do not discuss this aspect or it is tackled by directly adding the variables (Balteanu et al., 2015). The HSA enables the division of vulnerability variables into groups. However, it did not provide information on relative significance of variables within each group, making subsequent implementation of the PCA necessary (Cutter et al., 2003; Fekete, 2009; Cutter et al., 2013; Nelson et al., 2015; Hummell et al., 2016). The weighting method implemented here makes it possible to assign a different weight for each vulnerability factor, according to whether it provides more or less redundant information to the ISVI. Although many authors support the idea of assigning equal weight to the factors (Chakraborty et al., 2005), it seems reasonable to think that not all factors have the same importance in the ISVI construction (Brooks et al., 2005; Eakin and Luers, 2006; Liu and Li, 2016), especially when the number of variables that form each factor and their explained variance can vary. It is also possible that even the importance of each factor varies spatially. The above can be solved by carrying out geographically weighted principal component analysis (GWPCA) (Frazier et al., 2014; Gollini et al., 2015).

## 4.2 Integrated social vulnerability and variables involved

In spite of differences among variables considered in literature as a means of explaining social vulnerability, there are some key variables common to all considered indicators, such as age, gender, race, socioeconomic status and living conditions (Cutter et al., 2003; Adger et al., 2004; Penning-Rowsell et al., 2005; Frazier et al., 2014). However, each region presents particular characteristics and constraints, which must be considered during the variable selection procedure (Frazier et al.,



2014). Vulnerability factors identified in Castilla y León (see Table 2) reflect the specific characteristics of this region whose cartographic representation gives us an idea of the vulnerability spatial distribution and helps us spatially identify vulnerability hotspots (see Fig. 4). Vulnerability factors which make up the exposure component (see Fig. 4 (a), (b) and (d)) are mainly related to public buildings such as schools, kindergartens or health facilities. Usually they are occupied by
sensitive people (e.g., small children or elderly population, patients, etc.), who generally require external assistance during an evacuation due to flash floods. Moreover, single-family dwellings abound in the study area, which tend to have basements and where there are usually rooms on the ground floor (i.e., living rooms, kitchens and sometimes even bedrooms), being both spaces prone to flooding (Bodoque et al., 2016b; Karagiorgos et al., 2016).

Concerning vulnerability factors making up the sensitivity component (see Fig. 4 (e), (g), (h), (j) and (k)), urban areas of
interest have a mean dependency rate higher than 70%, particularly due to elderly people who hinder the population evacuation process because they tend to have reduced mobility. Moreover, the elderly usually need economic support in the post-disaster period (Cutter et al., 2003). Unemployment is another vulnerability factor to be considered. It is related to the possible inability of a household to invest economical resources in flood insurance or in flood mitigation measures, which contribute to a slower recovery (Cutter et al., 2003; Fekete, 2010). With regard to health facility accessibility, the usual lack
of nearby medical services in the urban areas studied may make immediate relief difficult and extend disaster recovery (Cutter et al., 2003).

Finally, as far as the resilience component is concerned (see Fig. 4 (c), (f) and (i)), households in good condition were considered to have high structural capacity to cope with flood impacts, so that direct losses and repair costs would be lower (Cutter et al., 2003). Inhabitants aged 15 to 64 were also deemed to be a resilient factor, since they can help to evacuate
population during a flash flood event (Fekete, 2010). Lastly, urban areas with a higher available public budget per capita may implement a larger number of mitigation measures aiming to reduce flood damage. Fixed investments per capita are related to the level of economic wealth, which can determine the ability to absorb losses and enhance resilience (e.g., by means of implementing individual flood risk mitigation measures) (Kunreuther et al., 2013; Haer et al., 2016).

Integrated social vulnerability assessment analyzes interactions among the different vulnerability components and even
between them and the ISVI (see Fig. 5). In addition, there is great heterogeneity in the combination of vulnerability components that generate the different ISVI categories. In spite of the above, the most vulnerable urban areas have the highest exposure component values. Urban areas in the high ISVI category usually have higher values for the sensitivity component than for exposure, although exposure quintile categories range from 2 to 5. Urban areas included in very low and low ISVI categories have the highest resilience component values, coinciding with the lowest levels of exposure. Thus, the
highest ISVI values are mainly controlled by the exposure component.

These variations in ISVI values confirm the idea supported by other authors that vulnerability has a high spatial variability and, therefore, it cannot be treated homogeneously (Cutter et al., 2008; Frazier et al., 2014). Integrated social vulnerability assessments not only help to know which factors should be acted on to reduce vulnerability, but also which factors should be strengthened to increase resilience. In the same way, identification of vulnerability patterns (see Fig. 6) also helps us discern

the sources of vulnerability and resilience within each cluster of urban areas and, especially, if these influences are direct or inverse and how strong they are. This facilitates the development of specific strategies of FRM for each cluster. For instance, cluster 2 is composed of urban areas with ageing populations, usually lacking nearby health facilities and whose households often are in poor condition. Therefore, specific mitigation measures could be proposed in these areas such as increasing

financial support to help families meet the cost of implementing individual flood risk mitigation measures or promoting renovation of households in poor condition through a system of economic incentives, as well as the suitable design of action and evacuation protocols that reduce potential losses derived from flash flood events, particularly with regard to elderly people with reduced mobility.

### 4.3 Policy implications

As high human and economic losses continue today due to flash floods (Wilhelmi and Morss, 2013), this draws attention to the need for a change in traditional FRM towards an integrated approach, which requires comprehensive analysis of the social risk component (Koks et al., 2015). For this purpose, it is essential to perform a social vulnerability analysis from a holistic point of view, which means that not only it is important to identify which socioeconomic and demography characteristics increase population sensitivity to be damaged by a flash flood, but also to know which features increase a

population's capacity to resist, cope with and recover from its impacts (Cutter et al., 2010; Frazier et al., 2014; Zhou et al., 2015), as has been done here. That would enable local competent authorities to plan and implement specific strategies to reduce vulnerability and strengthen resilience which also means developing specific mitigation measures to reduce flood risks (Frazier et al., 2014; Nelson et al., 2015; Hummell et al., 2016), going further than the traditional approach of delineation of flood prone areas and design of structural mitigation measures (i.e. seeking for not only reducing flood

hazard). However, in order to achieve greater effectiveness for FRM plans, it is necessary to engage all stakeholders in them, both public authorities and communities (Eakin and Luers, 2006; Koks et al., 2015; Haer et al., 2016). This is really important in small mountainous areas prone to flash flooding, because they are managed by local administrations where available economic resources tend to be limited, so individual adaptation measures are particularly relevant, partly depending on the risk perception and the awareness level (Bodoque et al., 2016a). Furthermore, both individual social

networks and social context are of key importance in decision-making related to public preparedness (Haer et al., 2016). Thus, and since the social component plays a decisive role, a suitable design is required for flood risk communication strategies accompanying integrated social vulnerability analysis. Traditional top-down communication strategies have proven ineffective, so a change is currently occurring towards people-centred strategies, which seeks to reflect population heterogeneity (Bodoque et al., 2016a; Haer et al., 2016). Therefore, a comprehensive characterization of the social

component of flood risk requires not only an integrated social vulnerability assessment, but also that affected population are aware of their situation and have appropriate knowledge to reduce possible flood impacts individually (Albano et al., 2015), thus seeking social learning which can be translated into a disaster risk reduction (Cutter et al., 2008).



## 5 Conclusions

A comprehensive characterization of social vulnerability is critical for an integrated FRM. To our knowledge, this is the first time that an integrated social vulnerability assessment has been performed in the context of flash floods. The implementation of an HSA helps to overcome PCA sample size limitation, meaning an alternative methodology to construct an ISVI in areas where available data is limited. Moreover, identification of vulnerability spatial patterns through the LCCA gives the sources of vulnerability in each urban area. Social vulnerability is not only characterized by any demographic and socioeconomic variables that condition a population's ability to cope with flood impacts, but also by any features which help the population to resist and recover from it. Thus, the integrated social vulnerability analysis is particularly important, since it considers exposure, sensitivity and resilience components in a holistic manner analyzing their interactions. Therefore, the approach implemented here seeks not only an improvement in the social vulnerability calculation, but also a better connection between conceptual and practical areas, which results in a more suitable development of FRM plans and policies and an increase in its efficiency.

**Author contribution**

Jose M. Bodoque and A. Diez-Herrero developed the idea. E. Aroca-Jimenez built the database. Juan A. Garcia designed the methodological process. Juan A. Garcia and E. Aroca-Jimenez carried it out. E. Aroca-Jimenez prepared the manuscript with contributions from all co-authors.

The authors declare that they have no conflict of interest.

**Acknowledgements**

This research has been funded by the MARCoNI Project (MINECO, CGL2013-42728-R) of the Spanish National Plan for Scientific and Technical Research and Innovation.

The authors wish to thank the following for their help: the Castilla y León regional government and regional employment observatory; Regional Civil Protection Department of National Government (José Luis González Álvarez, Roberto Martínez-Alegría López and volunteers); and finally all the municipal councils included in this study. The authors would also like to thank Dr. Susan Cutter, Hazards and Vulnerability Research Institute, University of South Carolina for her helpful comments on an earlier draft of the paper.



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
