# Peer review of "Construction of an Integrated Social Vulnerability Index in urban areas prone to flash flooding"

_Natural Hazards and Earth System Sciences, 2016_

## Referee Comment (RC1) · Anonymous Referee #1 · 12 Feb 2017

This paper aims to calculate an Integrated Social Vulnerability Index (ISVI) to flash flood processes in the region of Castilla y Leon in central Spain, considers exposure, sensitivity and resilience. To address the main objective, a set of variables were analyzed using hierarchical segmentation analysis and in the next level by performing a Principal Components Analysis. Tolerance statistic was used as a variable weighting method and in the last step, Latent Class Analysis was performed to identify social vulnerability profiles within the study area.

General comment

I have read the paper with great interest and the main objective addressed by the manuscript is framed appropriately to the scope of the journal, but there is some con-

fusion regarding to the term vulnerability. Therefore, I think that the paper needs some revisions and I recommend to accept it only after these revisions.

Specific comments

Introduction. In general, vulnerability in the context of natural hazards is a broad term, which covers different dimensions from physical to social approaches. In this line, it is important from the authors to give a clear framework of the vulnerability concept used in this study. Try to explain better or make more explicit the links what do you deal with. For example, it is not clear to me what the authors understand as vulnerability, integrated vulnerability and the components influencing vulnerability. In this part and in order to avoid confusion, I would suggest the authors to clearly indicate what they define as vulnerability in the context of the existing frameworks as well as a clear definition of the terms exposure, sensitivity and resilience.

Materials and Methods. The study area is well described. The methodological outline is well described and the method sounds scientifically correct (I am not an expert on statistics). I would suggest the authors to make figure 2 more simple by reducing some information that is presented on the text. Moreover, it is not entire clear to me, why the authors used a low probability scenario and not scenarios with medium or high probability. In page 5/line 6, I would recommend the authors to create a new subchapter with the database generation. Moreover, I would suggest them to describe a bit more the data used and to give some more information about the surveys done (i.e. telephone calls and/or personal research). Additionally, on the construction of the Integrated Social Vulnerability Index (part 2.2.4), I would recommend the authors to describe a bit more the idea behind the equation's modification from the original one presented by Frazier et al. (2014).

Results. In general, I would suggest the authors to describe only their results to this part and to remove some parts describing methods (i.e. page11/ lines 1-5 or page 12/ lines 5-6) on the methodology part as well as some parts discussing their results (i.e.

page 16/lines 11-13) to the discussion part. On figure 3, I would suggest to add the description of the variables to increase reader's friendliness.

At the end, the conclusions presented are too general and do not reflect what exactly shown in this study. Conclusions based on the findings of the analysis presented would be more effective.

───────────────────────

---

## Author Comment (AC1) · 20 Mar 2017

Dear,

I am submitting a revised copy of our manuscript "Construction of an Integrated Social Vulnerability Index in urban areas prone to flash flooding" (doi:10.5194/nhess-2016-408) by Aroca-Jimenez et al. We are very grateful to the reviewer for the helpful comments on our manuscript. We have addressed all the comments made by the reviewer. To facilitate the review, we have modified the manuscript highlighting in yellow the changes carried out (please see Supplement document). We have taken advantage of this new opportunity to improve text, figures and tables as the reviewer has requested. In this regard, the concept of both vulnerability and all its components

(i.e. sensitivity, exposure and resilience) have been clarified. As the reviewer recommended, we have created a new subsection under the section 2 (i.e. "2.2.2 Database generation"). Moreover, we have modified Figure 3 by adding the description of the variables in order to increase readers' friendliness as reviewer suggested. To facilitate understanding of the results, we have added a new column to Table 2, indicating the vulnerability component to which each vulnerability factor belongs. Conclusions have been amended to express clearer how the methodology proposed here constitutes an improvement on the state of the art and the extent to which the results may be included in flood risk management plans.

We thank you for the opportunity to resubmit our manuscript to the journal Natural Hazards and Earth System Sciences and hope that it is now suitable for publication. We look forward to hearing from you at your earliest convenience.

CITED REFERENCES: -Adger, W. N.: Vulnerability, Global Environmental Change-Human and Policy Dimensions, 16, 268-281, 10.1016/j.gloenvcha.2006.02.006, 2006. -Birkmann, J., Cardona, O. D., Carreno, M. L., Barbat, A. H., Pelling, M., Schneiderbauer, S., Kienberger, S., Keiler, M., Alexander, D., Zeil, P., and Welle, T.: Framing vulnerability, risk and societal responses: the MOVE framework, Natural Hazards, 67, 193-211, http://dx.doi.org/110.1007/s11069-11013-10558-11065, 10.1007/s11069-013-0558-5, 2013. -Cutter, S. L., Boruff, B. J., and Shirley, W. L.: Social vulnerability to environmental hazards, Social Science Quarterly, 84, 242-261, http://dx.doi.org/210.1111/1540-6237.8402002, 10.1111/1540-6237.8402002, 2003. -Frazier, T. G., Thompson, C. M., and Dezzani, R. J.: A framework for the development of the SERV model: A Spatially Explicit Resilience-Vulnerability model, Applied Geography, 51, 158-172, http://dx.doi.org/110.1016/j.apgeog.2014.1004.1004, 10.1016/j.apgeog.2014.04.004, 2014.

Please also note the supplement to this comment:
http://www.nat-hazards-earth-syst-sci-discuss.net/nhess-2016-408/nhess-2016-408-

[Figure]

AC1-supplement.pdf

[Figure]

| No. | Comment | Location in the submitted paper | Location in the reviewed paper | Amendment |
|---|---|---|---|---|
| | | | | **Responses to the reviewer 1' comments** |
| 1 | I would suggest the authors to clearly indicate what they define as vulnerability in the context of the existing frameworks as well as a clear definition of the terms exposure, sensitivity and resilience | Pages 1-3 (Introduction). | Page 2, Lines 15-17, and Page 3, Lines 2-3. | In agreement with the reviewer, we have included the theoretical concepts of vulnerability, sensitivity, exposure and resilience, in which the integrated social vulnerability index is based on. The key parameters or components of vulnerability are exposure, sensitivity and resilience (Adger, 2006; Birkmann et al., 2013). The social dimension of vulnerability (i.e. social vulnerability) has been traditionally estimated through the construction of an index, which is composed of several vulnerability factors (usually derived from a factor analysis or principal component analysis) (Cutter et al., 2003). Each vulnerability factor is in turn composed of several variables (variables considered as a means of explaining social vulnerability, such as age, gender, unemployment...). Traditional social vulnerability analysis usually shows the results for each vulnerability factor and for the total social vulnerability (i.e. the combination of the above vulnerability factors), but they do not analyze the results by component. In this regard, the main objective of an integrated social vulnerability analysis is to find out the involvement of each vulnerability component (i.e. sensitivity, exposure and resilience) to the total vulnerability (Frazier et al., 2014), which facilitates the incorporation of the analysis results into the flood risk management plans, particularly at regional scales. |
| 2 | It is not clear to me what the authors understand as vulnerability, integrated vulnerability and the components influencing vulnerability. | Pages 1-3 (Introduction). | Page 3, Lines 16-18. | We have clarified the concept of "integrated vulnerability analysis". This concept analyzes separately the different vulnerability components (i.e. exposure, sensitivity and resilience) and their involvement in total vulnerability, assessing, in addition, the interactions among them. |
| 3 | I would suggest the authors to make figure 2 more simple by reducing some information that is presented on the text. | Page 5. | | Figure 2 has not been summarized, as we have considered that its inclusion is essential in order to understand the methodology of the paper. The methodological procedures followed, especially the statistical analysis, are so complex and interrelated that they are very difficult to follow with a simple description in the text. Thus, Figure 2 in its current state allows: i) to display on a single graph the entire methodological process from data sources to final results; ii) to understand the sequence of statistical analysis in parallel to their reading; and iii) to understand the relationships among the different methodological steps and procedures. A simplification of the Figure 2 would result in readers feeling that there are unjustified breaks within the methodological procedure. Finally, Figure 2 enables to replicate our methodological procedure by other researchers and, therefore, to be contrasted. |

**Fig. 1.** Responses to the reviewer 1' comments_1

| 4 | It is not entire clear to me, why the authors used a low probability scenario and not scenarios with medium or high probability | Page 5, Line 13. | | We have used the scenario of low or exceptional probability (500-year flood) because it is the flood hazard zone that is the most comprehensive representation of urban areas that could be affected by flash floods at regional scale, according to the European Flood Directive. |
|---|---|---|---|---|
| 5 | I would recommend the authors to create a new subchapter with the database generation. | Page 5, Line 5. | Page 6, Line 8. | Done. Thank you for the recommendation. |
| 6 | I would suggest them to describe a bit more the data used and to give some more information about the surveys done (i.e. telephone calls and/or personal research) | Page 6, Lines 6-11 | Page 6, Lines 12-14 | Done. Thank you for the suggestion. |
| 7 | I would recommend the authors to describe a bit more the idea behind the equation's modification from the original one presented by Frazier et al. (2014) | Page 9, Lines 8-9 and 11-12. | Page 9, Lines 8, 11 and 15-17. | We have clarified the modifications made in the equations presented by Frazier et al. (2014). So, we have replaced in the text the term "modified" by "adapted", since what we did was to adapt the equations to our terminology (i.e. changing the term "adaptive capacity" to "resilience"; Equation 1). In addition, we have used a different method to weight the vulnerability factors; Equation 2). A clarification in the text about the adaptation from Frazier et al. (2014) (lines 15-17) have been added. |
| 8 | I would suggest the authors to describe only their results to this part and to remove some parts describing methods on the methodology part as well as some parts discussing their results to the discussion part | Page 11, Lines 1-5 | Page 11, Line 1 | We have removed the text related methodology. |
| | | Page 12, Lines 5-6 | Page 13, Bottom of Figure 4 | We have moved the text to the bottom of Figure 4 since it was describing this picture. |
| | | Page 16, Lines 11-13 | | Done. As the reviewer recommends, we have eliminated certain parts of the text of the section 3.2 ("Social vulnerability patterns") because they were related to discussion of results. |
| 9 | I would suggest to add the description of the variables to increase reader's friendliness | Figure 3 | | Done. Thank you for the suggestion. |
| 10 | The conclusions presented are too general and do not reflect what exactly shown in this study. Conclusions based on the findings of the analysis presented would be more effective | Page 21, Lines 2-12 | Page 21, Lines 2-14 | We have reworded the conclusions trying to make them more specific. So, conclusions have been amended to express clearer how the methodology proposed here constitutes an improvement on the state of the art and the extent to which the results may be included in flood risk management plans. |

**Fig. 2.** Responses to the reviewer 1' comments_2

**Supplement:**

[revised manuscript text omitted]

---

## Referee Comment (RC2) · Anonymous Referee #2 · 10 Apr 2017

First of all, I very much enjoyed reading the manuscript. I have, however, a few comments to improve the manuscript. Please find them below.

- Abstract, line 16: 'it has not yet provided'. Please rephrase this a bit, the sentence is unclear.

- some additional explanation is required on the inclusion of exposure in the social vulnerability index. In the traditional risk framework, exposure and vulnerability are two different components of the framework. As many researchers from the risk field read this journal, it should be specifically emphasized that including exposure is common practice in the social vulnerability field, even though this may contradict to the definition of risk and vulnerability which is more commonly used in the disaster risk community.

This is important for the interpretation of the results.

- I have a few questions and a suggestion regarding Figure 2: - why is there an arrow going from Flash flood low probability municipalities to socioeconomic variables? Because the flash flood box is blue, it now seems like a hazard variable is added to the socioeconomic variables. This is, however, not the case (and should not be the case either). - why are sensitivity and exposure 'clustered' and is resilience not in this cluster? - Perhaps add a third colour that specifies the (final) results. This would make it more clear why some arrows exist in the framework (for instance the arrow from factor scores to clusters of municipalities).

- Section 2.2.2: I do not fully understand the use of the Euclidian distance method. If I do understand it correctly, the sum of the differences between variable values is considered to be the distance? So distance is not spatial? I think it would be good to explain this a bit more clearly, as some parts of the paper are spatial (the clusters of municipalities for instance). This causes (at least for me) some confusion.

- Captions of Figure 4 and Figure 5 could be a bit longer. Figure + figure caption should be self-explanatory.

- Figure 3 is perhaps not required, as it shows roughly the same as table 2? Perhaps move to appendix, as table 2 shows everything we would like to know (the variable clusters and the factor names)

- Section 3.2: I am a bit puzzled with the notion of 'optimum number of clusters'. What does an optimum amount of clusters mean? Ok the statistics say so, but as a practitioner, what would it matter if you would have four clusters? How would this change the interpretation of the results?

- Section 4.1: I would suggest to move parts of this to the method section. Most parts of this section are regarding the interpretation of the results. It is better to make this clear before the results section, instead of afterwards. A discussion after the results,

weakens, in my opinion, the results.

- Section 4.3: I suppose the clustering of municipalities is interesting from a policy making perspective. It would be good to link the clustering to this section. How can it improve policy making if we can identify similar municipalities?

- Please make the conclusions a bit more specific for this paper. What can we really learn from this paper, especially from a policy making perspective. What does this paper add, besides being the first study on flash floods? A few lines on the conclusions for the study region (specific patterns identified) would be interesting as well.

---

## Author Comment (AC2) · 27 Apr 2017

Dear, I am submitting a revised copy of our manuscript "Construction of an Integrated Social Vulnerability Index in urban areas prone to flash flooding" (doi:10.5194/nhess-2016-408) by Aroca-Jimenez et al. We are very grateful to the reviewer for the helpful comments on our manuscript. We have addressed all the comments made by the reviewer. To facilitate the review, we have modified the manuscript highlighting in green the changes carried out. Changes made after receiving the comments from the other reviewer are still highlighted in yellow. We have taken advantage of this new opportunity to improve text and figures as the reviewer has requested. In this regard, the inclusion of exposure in the Integrated Social Vulnerability Index has been

clarified, adding a new reference for this purpose. As the reviewer recommended, we have simplified the Figure 2 in order to improve its understanding. Moreover, we have explained better the concept of 'optimum number of clusters' at the results section, extending the information with regard to the BIC and the CAIC statistics. We have also modified the text of the section 4.3 ('Policy implications'), giving some practical examples of specific mitigation measures that can be suggested for each cluster of urban areas. Finally, conclusions have been amended to express clearer how the methodology proposed here constitutes an improvement on the state of the art and the extent to which the results may be included in flood risk management plans, as both reviewers have recommended. We thank you for the opportunity to resubmit our manuscript to the journal Natural Hazards and Earth System Sciences and hope that it is now suitable for publication. We look forward to hearing from you at your earliest convenience.

Please also note the supplement to this comment:
http://www.nat-hazards-earth-syst-sci-discuss.net/nhess-2016-408/nhess-2016-408-AC2-supplement.pdf
* * *
[Figure]

| Responses to the reviewer 2' comments | | | | |
|---|---|---|---|---|
| No. | Comment | Location in the submitted paper | Location in the reviewed paper | Amendment |
| 1 | Abstract, line 16: 'it has not yet provided'. Please rephrase this a bit, the sentence is unclear. | Line 16 | Lines 16-18. | We have rephrased the sentence. As the reviewer pointed out, the sentence was badly phrased. |
| 2 | Some additional explanation is required on the inclusion of exposure in the social vulnerability index. In the traditional risk framework, exposure and vulnerability are two different components of the framework. As many researchers from the risk field read this journal, it should be specifically emphasized that including exposure is common practice in the social vulnerability field, even though this may contradict to the definition of risk and vulnerability which is more commonly used in the disaster risk community. This is important for the interpretation of the results. | Pages 1-3 (Introduction) | Page 3, lines 7-9. | We have explained that the inclusion of exposure in the social vulnerability analysis is a common practice, as the reviewer recommended. We have also included a new reference in order to strength this idea. |
| 3 | I have a few questions and a suggestion regarding Figure 2:

- why is there an arrow going from Flash flood low probability municipalities to socioeconomic variables? Because the flash flood box is blue, it now seems like a hazard variable is added to the socioeconomic variables. This is, however, not the case (and should not be the case either).

- why are sensitivity and exposure 'clustered' and is resilience not in this cluster? | Figure 2 | Figure 2 | Overall, we have simplified the Figure 2 in order to make it clearer and easier to understand. Moreover, we have done a terminological change from 'municipalities' to 'urban areas', since municipalities is usually used to refer to administrative boundaries or local administration (i.e. the council). Thus, we have used the term 'urban areas' when we talk about the areas prone to flash flooding and 'municipalities' when we refer to the town halls.

- We wanted to show with this arrow that socio-economic variables had only been gathered to those urban areas that met the defined requirements, which were then named as 'Flash flood low probability municipalities'. In order to facilitate the understanding of this part of the Figure 2, we have modified the color of this box from a blue plain to a gradient blue-beige color, indicating that the 'Flash flood low probability municipalities' box is the beginning of the second part of the figure (beige boxes).

- All vulnerability components (exposure, sensitivity and resilience) were taken into account in the Latent Class Cluster Analysis (LCCA), using as input data the factor scores of the urban areas of interest. The minus sign which is placed on the arrow that goes from the box 'Factor scores' to 'Resilience' box |

**Fig. 1.**

| | | | | |
|---|---|---|---|---|
| 5 | - Perhaps add a third colour that specifies the (final) results. This would make it more clear why some arrows exist in the framework (for instance the arrow from factor scores to clusters of municipalities. | | | indicates the sign of the component when the Integrated Social Vulnerability Index (ISVI) was calculated, and not that Resilience was not considered in the LCCA. Traditionally, factors that express sensitivity or exposure are considered as positive values in the ISVI (see the plus sign that is placed on the arrow that go from the box 'Factor scores' to 'Sensitivity' and 'Exposure' boxes); while factors that state resilience are considered as negative values, as has been done here.

- Done. Thank you for the recommendation. |
| 4 | Section 2.2.2: I do not fully understand the use of the Euclidian distance method. If I do understand it correctly, the sum of the differences between variable values is considered to be the distance? So distance is not spatial? I think it would be good to explain this a bit more clearly, as some parts of the paper are spatial (the clusters of municipalities for instance). This causes (at least for me) some confusion. | Section 2.2.2 | Section 2.2.3. Page 8, Line 9. | - We have changed the term 'distance' by 'similarity' in the text in order to clarify that we were not talking about spatial distance. From a statistical point of view, distance measures are a type of similarity measure, so it is correct to use the term 'similarity'. We have used the Euclidean distance to evaluate how similar to each other the variables were, as is explained by Euclidean distance definition (page 8, lines 9-10). The greater the distance among variables are, the more similar the variables among them are. Hierarchical Segmentation Analysis (HSA) groups variables according to their similarity rather than the distance, that is why LCCA was implemented for. |
| 5 | Captions of Figure 4 and Figure 5could be a bit longer. Figure + figure caption should be self-explanatory. | Figures 4 and 5 | Figures 4 and 5 | We decided to shorten captions of Figure 4 and 5 in order to have a more balanced length of the text of the section 3.1. |
| 6 | Figure 3 is perhaps not required, as it shows roughly the same as table 2? Perhaps move to appendix, as table 2 shows everything we would like to know (the variable clusters and the factor names) | Figure 3 | Figure 3 | Figure 3 and Table 2 do not show the same information. Figure 3 corresponds to the Hierarchical Segmentation Analysis (HSA) output, while Table 2 corresponds to the Factor Analysis output, which includes factor loadings that are necessary to construct the ISVI. HSA helps to overcome the Principal Components Analysis sample size limitations, so we think that including the Dendrogram into the results section of this paper is an interesting development. |
| 7 | Section 3.2: I am a bit puzzle with the notion of 'optimum number of clusters'. What does an optimum amount of clusters mean? Ok the statistics say so, but as a practitioner, what would | Page 15, line 3 | Page 15, lines 2-3 | We have extended the information about the BIC and the CAIC criteria in the text. |

**Fig. 2.**

| | | | | |
|---|---|---|---|---|
| | it matter if you would have four clusters? How would this change the interpretation of the results? | | | BIC and CAIC are statistics that enable to establish a number of clusters, which can be used in flood management. We used the BIC and the CAIC criteria in order to select the more parsimonious number of clusters (i.e. the number of clusters that provides as much information as possible taking into account the number of parameters to estimate). The more information is explained by the model, the greater the number of estimated statistics will be. The above is shown in Table 3 (Page 15). The minimum values of the BIC and the CAIC statistics are reached with a model of 3 clusters, and their values increase again in the estimations that consider four and five clusters. This means that the number of parameters to estimate by the model (see Table 3, 'Number of parameters' column) are too high in comparison to the obtained increase of explained information ('Log-likelihood(LL)' column). From a practical point of view, the above means that an increase in the number of clusters from 3 to 4 or 5 would split a fairly homogeneous cluster of urban areas into several subgroups which would not be very different from each other. Therefore, a greater level of disaggregation would not help to improve the implementation of different flood risk mitigation measures for each cluster of urban areas. |
| 8 | Section 4.1: I would suggest to move parts of this to the method section. Most parts of this section are regarding the interpretation of the results. It is better to make this clear before the results section, instead of afterwards. A discussion after the results, weakens, in my opinion, the results. | Section 4.1, Page 18, Lines 22-23 | Section 4.1 | We have moved some parts of the text from section 4.1 to methodology section. |
| 9 | Section 4.3: I suppose the clustering of municipalities is interesting from a policy making perspective. It would be good to link the clustering to this section. How can it improve policy making if we can identify similar municipalities? | | | We have extended the text of this section including practical examples of specific mitigation measures to each cluster of urban areas. |
| 10 | Please make the conclusions a bit more specific for this paper. What can we really learn from this paper, especially from a policy making perspective. What does this paper add, besides being the first study on flash floods? A few lines on the conclusions for the study region (specific patterns identified) would be interesting as well. | Section 5 | Section 5 | We rewrote the conclusions after reviewer 1 recommendation. Conclusions were amended to express clearer how the methodology proposed here constitutes an improvement on the state of the art and the extent to which the results may be included in flood risk management plans. |

**Fig. 3.**

**Supplement:**

[revised manuscript text omitted]

---

## Author Comment (AC3) · 17 May 2017

Dear Referee #1,

We are very grateful for your helpful comments on our manuscript.

In spite of some confusion around the use of the vulnerability terminology, there is a certain consensus about what issues should be assessed to its characterization. The vulnerability analysis carried out in this paper has followed a hybrid approach (Eakin and Luers, 2006) between risk-hazard approaches, which considers that vulnerability depends on the biophysical risk factors and the potential loss of a particular exposed population (e.g. the hazards-of-place model of vulnerability (Cutter, 1996); and politi-

cal economy/political ecology approaches, which emphasize the political, cultural and socioeconomic factors that explain the differential exposure, impacts and capacities to recover from an impact (e.g. the pressure and release model (Blaikie et al., 1994). Taking into account the key parameters for the vulnerability research that highlight the above-mentioned approaches, we understand that vulnerability depends on the social system's exposure and sensitivity to stress (exposure and sensitivity components of our Integrated Social Vulnerability Index, ISVI) as well as its capacity to absorb or cope with the effects of these stressors (resilience component of our ISVI) (Eakin and Luers, 2006;Adger, 2006;Birkmann et al., 2013). In this context, we define 'exposure' as the people and assets susceptible to be harmed; 'sensitivity' as the level to which people and assets can be damaged; and 'resilience' as the ability to absorb, cope with and recover from the effects of a disaster.

Furthermore, the social dimension of vulnerability (i.e. social vulnerability) has been traditionally estimated through the construction of indexes, which are composed of several vulnerability factors (usually derived from a factor analysis or principal component analysis)(Cutter et al., 2003). Each vulnerability factor is in turn composed of several variables (variables considered as a means of explaining social vulnerability, such as age, gender, unemployment...). Traditional social vulnerability analysis usually shows the results for each vulnerability factor and for the total social vulnerability (i.e. the combination of the above vulnerability factors), but they do not analyze the results by component. We have constructed a social vulnerability index using an integrating approach (i.e. integrating elements from risk-hazard and political economy/political ecology approaches)(Eakin and Luers, 2006), which has been called Integrated Social Vulnerability Index (ISVI). This enables us to find out the involvement of each vulnerability component (i.e. sensitivity, exposure and resilience) to the total vulnerability and their interactions (Frazier et al., 2014), which also facilitates the incorporation of the analysis results into the flood risk management plans, particularly at regional scales.

We are aware of the complexity of the methodology section, so we appreciate your

comments about Figure 2. We will try to simplify Figure 2 in order to make it clearer, easier to understand and to avoid misunderstandings.

With regard to the identification of urban areas prone to flash flooding, we have used the scenario of low or exceptional probability (500-year flood) because it is the flood hazard zone that is the most comprehensive representation of urban areas that could be affected by flash floods at regional scale, according to the European Flood Directive (Directive 2007/60/EC of the European Parliament and of the Council of 23 October 2007 on the assessment and management of flood risks).

In agreement with the reviewer, we will create a new subsection with the database generation. Moreover, we will extend this subsection including more information about the variables included and how they were gathered, as the reviewer recommend.

Regarding the comment about further describing the idea behind the equation's modification from the original one presented by Frazier et al. (2014), this author also used an integrated approach in the development of their Spatially Explicit Resilience-Vulnerability (SERV) model. However, the equations used in our ISVI represent an adaptation from the ones used in the above-mentioned article, since we have adapted the equations to our terminology (i.e. changing the term 'adaptive capacity' to 'resilience') and we have used a different method to weigh the vulnerability factors (i.e. using tolerance statistic instead of the percentage of explained variance).

As the reviewer recommend, we will review the text of the results section in order to remove those parts that describe methodology or discussion.

We agree with the reviewer that by including the description of the variables on Figure 3 this would increase reader's friendliness, so Figure 3 will be modified in the revised draft of the manuscript.

Finally, we appreciate the comments about the conclusions. They will be amended in order to express clearer how the methodology proposed here constitutes an improvement on the state of the art and the extent to which the results may be included in flood risk management plans and therefore improve flood risk management, which is the main objective of this social vulnerability analysis.

Best regards,

Estefania Aroca-Jiménez.

CITED REFERENCES:

Adger, W. N.: Vulnerability, Global Environmental Change-Human and Policy Dimensions, 16, 268-281, 10.1016/j.gloenvcha.2006.02.006, 2006. Birkmann, J., Cardona, O. D., Carreno, M. L., Barbat, A. H., Pelling, M., Schneiderbauer, S., Kienberger, S., Keiler, M., Alexander, D., Zeil, P., and Welle, T.: Framing vulnerability, risk and societal responses: the MOVE framework, Natural Hazards, 67, 193-211, http://dx.doi.org/110.1007/s11069-11013-10558-11065, 10.1007/s11069-013-0558-5, 2013. Blaikie, P., Cannon, T., Davis, I., and Wisner, B.: At risk: natural hazards, people's vulnerability, and disasters, Routledge, London, 304 pp., 1994. Cutter, S. L.: Vulnerability to environmental hazards, Progress in Human Geography, 20, 529-539, 10.1177/030913259602000407, 1996. Cutter, S. L., Boruff, B. J., and Shirley, W. L.: Social vulnerability to environmental hazards, Social Science Quarterly, 84, 242-261, http://dx.doi.org/210.1111/1540-6237.8402002, 10.1111/1540-6237.8402002, 2003. Eakin, H., and Luers, A. L.: Assessing the vulnerability of social-environmental systems, Annual Review of Environment and Resources, 31, 365-394, http://dx.doi.org/310.1146/annurev.energy.1130.050504.144352, 10.1146/annurev.energy.30.050504.144352, 2006. Frazier, T. G., Thompson, C. M., and Dezzani, R. J.: A framework for the development of the SERV model: A Spatially Explicit Resilience-Vulnerability model, Applied Geography, 51, 158-172, http://dx.doi.org/110.1016/j.apgeog.2014.1004.1004, 10.1016/j.apgeog.2014.04.004, 2014.

---

## Author Comment (AC4) · 17 May 2017

Dear Referee #2,

We are most grateful for your helpful comments on our manuscript.

We appreciate the grammatical comments, and we will further review the text of the manuscript.

As you point out, exposure is commonly included in the social vulnerability analysis. Although exposure and vulnerability are two different components of risk, currently exposure is included in the social vulnerability assessments in order to provide a holistic characterization of vulnerability, since it is not possible to talk about potential for loss
(i.e. vulnerability) in the absence of exposure (Frazier et al., 2014). In the same way, resilience is also included in social vulnerability analysis, since the potential for loss also depends on the ability to absorb, cope with and recover from the effects of a disaster. Thus, a comprehensive social vulnerability assessment should include the social system's exposure and sensitivity to stress (exposure and sensitivity components of our Integrated Social Vulnerability Index, ISVI) as well as its capacity to absorb or cope with the effects of these stressors (resilience component of our ISVI) (Eakin and Luers, 2006). Therefore, we will clarify this point explaining in the revised version of the manuscript that the inclusion of exposure in social vulnerability analysis is a common practice in the social vulnerability field as you recommend.

Considering the comments received by both reviewers about Figure 2, we will try to simplify it in order to make it clearer and easier to understand. We will take advantage of the opportunity to homogenize the use of 'municipalities' and 'urban areas' terms. As you recommend, we will add a third color in Figure 2 in order to show the final results with more clarity. We appreciate your comments about Figure 2 and we will review its design in order to avoid misunderstandings.

The Euclidean distance is a type of distance measure, but it is not a spatial distance. From a statistical point of view, distance measures belong to the so-called similarity measures. We have used the Euclidean distance to evaluate how similar to each other the variables were, as the Euclidean distance definition included in the paper explains. The greater the distance among variables, the less similar the variables are. Therefore, the Hierarchical Segmentation Analysis (HSA) groups the variables according to their similarity rather than the spatial distance.

We understand your comment about captions of Figure 4 and Figure 5, but we decided to shorten them in order to have a more balanced length of the text of this section. On the other hand, Figure 3 and Table 2 do not show the same information. Figure 3 corresponds to HSA output while Table 2 corresponds to the Factor Analysis (FA) output, which includes factor loadings that are necessary to construct the ISVI. HSA

helps to overcome the Principal Components Analysis (PCA) sample size limitations, so we think that including the HSA output (i.e. the Dendrogram) into the results section of this paper is an interesting approach.

As you recommend, we will extend the information about the BIC and the CAIC criteria in the text. BIC and CAIC are two statistics that enable to establish the optimum number of clusters in which the urban areas considered can be divided, which can be used in flood management. We used the BIC and the CAIC criteria in order to select the more parsimonious number of clusters (i.e. the number of clusters that provides as much information as possible taking into account the number of parameters to estimate by the model). The more information that is explained by the model, the greater the number of estimated parameters will be. In our case, the minimum values of the BIC and the CAIC statistics are reached with a model of 3 clusters of urban areas, which marks the point in which an increase in the number of clusters and therefore in the number of estimated parameters by the model do not represent a significant increase in the explained information. From a practical point of view, the above means that a greater level of disaggregation from the considered as optimum (i.e. considering 3 clusters of urban areas) would split a fairly homogeneous cluster of urban areas into several subgroups which would not be very different from each other, which would not help to improve the implementation of different flood risk mitigation measures for each cluster of urban areas.

We appreciate your comment about the discussion section. We will review the text and we will try to move those parts that interpret the results from this section to the methodology section in order to strengthen the results of the paper.

As you recommend, we will extend the text of the 'policy implications' section trying to emphasize the practical utility from a policy making perspective of the Latent Class Cluster Analysis (LCCA). For this purpose, we will include practical examples of specific mitigation measures that can be proposed for each cluster of urban areas identified by the LCCA.

Finally, we thank you for your comments about the conclusions. They will be amended in order to express clearer how the methodology proposed here constitutes an improvement on the state of the art and the extent to which the results may be included in flood risk management plans and therefore improve flood risk management, which is the main objective of this social vulnerability analysis.

Best regards,

Estefania Aroca-Jiménez.

CITED REFERENCES:

Eakin, H., and Luers, A. L.: Assessing the vulnerability of social-environmental systems, Annual Review of Environment and Resources, 31, 365-394, http://dx.doi.org/310.1146/annurev.energy.1130.050504.144352, 10.1146/annurev.energy.30.050504.144352, 2006. Frazier, T. G., Thompson, C. M., and Dezzani, R. J.: A framework for the development of the SERV model: A Spatially Explicit Resilience-Vulnerability model, Applied Geography, 51, 158-172, http://dx.doi.org/110.1016/j.apgeog.2014.1004.1004, 10.1016/j.apgeog.2014.04.004, 2014.

---

## Author Comment (AC5) · 23 May 2017

We are very grateful to the reviewer for the helpful comments on our manuscript. We have addressed all the comments made by the reviewer. To facilitate the review, we have modified the manuscript highlighting in yellow the changes carried out. We have taken advantage of this new opportunity to improve text, figures and tables as the reviewer has requested. In this regard, the concept of both vulnerability and all its components (i.e. sensitivity, exposure and resilience) have been clarified. As the reviewer recommended, we have created a new subsection under the section 2 (i.e. "2.2.2 Database generation"). Moreover, we have modified Figure 3 by adding the description of the variables in order to increase readers' friendliness as the reviewer suggested.

To facilitate the understanding of the results, we have added a new column to Table 2, indicating the vulnerability component to which each vulnerability factor belongs. Conclusions have been amended to express clearer how the methodology proposed here constitutes an improvement on the state of the art and the extent to which the results may be included in flood risk management plans.

Reply to Anonymous Referee #1 comments and changes made

I have read the paper with great interest and the main objective addressed by the manuscript is framed appropriately to the scope of the journal, but there is some confusion regarding to the term vulnerability. Therefore, I think that the paper needs some revisions and I recommend to accept it only after these revisions.

Specific comments

Introduction.

Comment 1 - In general, vulnerability in the context of natural hazards is a broad term, which covers different dimensions from physical to social approaches. In this line, it is important from the authors to give a clear framework of the vulnerability concept used in this study. Try to explain better or make more explicit the links what do you deal with. For example, it is not clear to me what the authors understand as vulnerability, integrated vulnerability and the components influencing vulnerability. In this part and in order to avoid confusion, I would suggest the authors to clearly indicate what they define as vulnerability in the context of the existing frameworks as well as a clear definition of the terms exposure, sensitivity and resilience.

Reply 1 - In spite of some confusion around the use of the vulnerability terminology, there is a certain consensus about what issues should be assessed to its characterization. The vulnerability analysis carried out in this paper has followed a hybrid approach (Eakin and Luers, 2006) between risk-hazard approaches, which considers that vulnerability depends on the biophysical risk factors and the potential loss of a particular

exposed population (e.g. the hazards-of-place model of vulnerability; Cutter, 1996); and political economy/political ecology approaches, which emphasize the political, cultural and socioeconomic factors that explain the differential exposure, impacts and capacities to recover from an impact (e.g. the pressure and release model; Blaikie et al., 1994). Taking into account the key parameters for the vulnerability research that highlight the above-mentioned approaches, we understand that vulnerability depends on the social system's exposure and sensitivity to stress (exposure and sensitivity components of our Integrated Social Vulnerability Index, ISVI) as well as its capacity to absorb or cope with the effects of these stressors (resilience component of our ISVI) (Eakin and Luers, 2006; Adger, 2006; Birkmann et al., 2013). In this context, we define 'exposure' as the people and assets susceptible to be harmed; 'sensitivity' as the level to which people and assets can be damaged; and 'resilience' as the ability to absorb, cope with and recover from the effects of a disaster. Furthermore, the social dimension of vulnerability (i.e. social vulnerability) has been traditionally estimated through the construction of indexes, which are composed of several vulnerability factors (usually derived from a factor analysis or principal component analysis)(Cutter et al., 2003). Each vulnerability factor is in turn composed of several variables (variables considered as a means of explaining social vulnerability, such as age, gender, unemployment...). Traditional social vulnerability analysis usually shows the results for each vulnerability factor and for the total social vulnerability (i.e. the combination of the above vulnerability factors), but they do not analyze the results by component. We have constructed a social vulnerability index using an integrating approach (i.e. integrating elements from risk-hazard and political economy/political ecology approaches)(Eakin and Luers, 2006), which has been called Integrated Social Vulnerability Index (ISVI). This enables us to find out the involvement of each vulnerability component (i.e. sensitivity, exposure and resilience) to the total vulnerability and their interactions (Frazier et al., 2014), which also facilitates the incorporation of the analysis results into the flood risk management plans, particularly at regional scales.

Change 1 - In view of the above, and in agreement with the reviewer, we have included

in the text the theoretical concepts of: - vulnerability assessment (page 2, lines 15-17). "Many efforts were put in flood hazard analysis in past, but vulnerability assessment (i.e. the analysis of the characteristics of a person or group and their situation that influence their capacity to anticipate, cope with, resist and recover from the impact of a natural hazard) is still one of the biggest constraints in flood risk assessment to date". - sensitivity, exposure and resilience, in which the integrated social vulnerability index is based on (page 3, lines 2-3). "Less attention has been paid to integrated analysis of vulnerability components, which considers the differential influence of exposure (i.e. people and assets susceptible to be harmed), sensitivity (i.e. the level to which people and assets can be damaged) and resilience (i.e. the ability to absorb, cope with and recover from the effects of a disaster) on total vulnerability". - integrated vulnerability (page 3, lines 16-18). "This paper aims to calculate an integrated social vulnerability index (ISVI) to flash floods, which considers separately the vulnerability components (i.e. exposure, sensitivity and resilience), analyzing the involvement of each of them in total vulnerability".

Materials and Methods.

Comment 2 - The study area is well described. The methodological outline is well described and the method sounds scientifically correct (I am not an expert on statistics). I would suggest the authors to make figure 2 more simple by reducing some information that is presented on the text.

Reply 2 - We have simplified the Figure 2 in order to make it clearer and easier to understand. Moreover, we have done a terminological change from 'municipalities' to 'urban areas', since municipalities is usually used to refer to administrative boundaries or local administration (i.e. the council). Thus, we have used the term 'urban areas' when we talk about the areas prone to flash flooding and 'municipalities' when we refer to the town halls. Moreover, we have added the green color in order to show clearer the final results.

Comment 3 - Moreover, it is not entire clear to me, why the authors used a low probability scenario and not scenarios with medium or high probability.

Reply 3 - We have used the scenario of low or exceptional probability (500-year flood) because it is the flood hazard zone that is the most comprehensive representation of urban areas that could be affected by flash floods at regional scale, according to the European Flood Directive.

Comment 4 - In page 5/line 6, I would recommend the authors to create a new subchapter with the database generation.

Change 4 - Done (page 6, line 8). Thank you for the recommendation. "2.2.2 Database generation"

Comment 5 - Moreover, I would suggest them to describe a bit more the data used and to give some more information about the surveys done (i.e. telephone calls and/or personal research).

Reply 5 - We have extended the 'Database generation' subsection including more information about the variables included in the integrated social vulnerability analysis and how they were gathered (page 6, lines 12-14).

Change 5 - "However, the other 29 variables were requested from certain public organizations or councils by means of telephone calls, asking for information directly from the person in charge (e.g., dependency, development and infrastructures...) or generated through personal research estimating the variables through other non-specific sources of information (e.g., collective vulnerability, healthcare services...)".

Comment 6 - Additionally, on the construction of the Integrated Social Vulnerability Index (part 2.2.4), I would recommend the authors to describe a bit more the idea behind the equation's modification from the original one presented by Frazier et al. (2014).

Reply 6 - Frazier et al. (2014) also used an integrated approach in the development of

their Spatially Explicit Resilience-Vulnerability (SERV) model. However, the equations used in our ISVI represent an adaptation from the ones used in the above-mentioned article, since we have adapted the equations to our terminology (i.e. changing the term 'adaptive capacity' to 'resilience') and we have used a different method to weigh the vulnerability factors (i.e. using tolerance statistic instead of the percentage of explained variance). Therefore, we have replaced in the text the term 'modified' by 'adapted' (page 9, lines 8 and 11) and we have added a clarification in the text about this adaptation (page 9, lines 15-17).

Change 6 - "The index construction method implemented coincides with the same method developed by Frazier et al. (2014), although the tolerance statistic was used here as a weighting method".

Results.

Comment 7 - In general, I would suggest the authors to describe only their results to this part and to remove some parts describing methods (i.e. page11/ lines 1-5 or page 12/lines 5-6) on the methodology part as well as some parts discussing their results (i.e. page 16/lines 11-13) to the discussion part.

Reply 7 - We have removed the text related to the methodology and the discussion from the results section. Moreover, we have moved the text of page 12 (lines 5-6) to the bottom of Figure 4 since it was describing this picture (page 13, bottom of Figure 4).

Change 7 - "Figure 4: Factor scores for identified vulnerability factors. For exposure and sensitivity factors, very high categories correspond to red colors while for resilience factors, very high categories correspond to blue colors".

Comment 8 - On figure 3, I would suggest to add the description of the variables to increase reader's friendliness.

Reply 8 - Figure 3 have been modified including the description of the variables. Thank

you for the suggestion.

Comment 9 - At the end, the conclusions presented are too general and do not reflect what exactly shown in this study. Conclusions based on the findings of the analysis presented would be more effective.

Reply 9 - The conclusions have been reworded trying to make them more specific. They have been amended in order to express clearer how the methodology proposed here constitutes an improvement on the state of the art and the extent to which the results may be included in flood risk management plans and therefore improve flood risk management, which is the main objective of this social vulnerability analysis (page 21, lines 2-14).

Change 9 - "A comprehensive characterization of social vulnerability is critical for an integrated FRM. The implementation of an HSA helps to overcome PCA sample size limitation, meaning an alternative methodology to the usually used to construct an ISVI in areas where available data is limited. The results show the high spatial heterogeneity of the social vulnerability within the study region and the high variability in the ISVI scores regarding the interactions between vulnerability components, which 5 makes an integrated analysis more important. The identification of vulnerability patterns through the LCCA gives the sources of vulnerability in each urban area, which simplifies the spatial heterogeneity analysis of the social vulnerability and enables to know what aspects need to be improved in order to decrease sensitivity and exposure (e.g. urban areas that compose cluster 2 are mainly made up of elderly people that usually need an external aid to reach shelter, so they should develop very effective evacuation plans in order to coordinate the different competent authorities) and what aspects need to 10 be reinforced to increase resilience (e.g. a high percentage of dwellings of urban areas that compose cluster 2 are in poor condition, so they could reverse this situation providing financial aid or promoting a tax cut of dwellings in good condition in order to involve population). Thus, a better integration of the ISVI results into FRM plans and policies is made possible enabling to propose specific strategies of vulnerability

reduction, increasing their efficiency".

CITED REFERENCES:

Adger, W. N.: Vulnerability, Global Environmental Change-Human and Policy Dimensions, 16, 268-281, 10.1016/j.gloenvcha.2006.02.006, 2006.

Birkmann, J., Cardona, O. D., Carreno, M. L., Barbat, A. H., Pelling, M., Schneiderbauer, S., Kienberger, S., Keiler, M., Alexander, D., Zeil, P., and Welle, T.: Framing vulnerability, risk and societal responses: the MOVE framework, Natural Hazards, 67, 193-211, 10.1007/s11069-013-0558-5, 2013.

Blaikie, P., Cannon, T., Davis, I., and Wisner, B.: At risk: natural hazards, people's vulnerability, and disasters, Routledge, London, 304 pp., 1994.

Cutter, S. L.: Vulnerability to environmental hazards, Progress in Human Geography, 20, 529-539, 10.1177/030913259602000407, 1996.

Cutter, S. L., Boruff, B. J., and Shirley, W. L.: Social vulnerability to environmental hazards, Social Science Quarterly, 84, 242-261, 10.1111/1540-6237.8402002, 2003.

Eakin, H., and Luers, A. L.: Assessing the vulnerability of social-environmental systems, Annual Review of Environment and Resources, 31, 365-394, 10.1146/annurev.energy.30.050504.144352, 2006.

Frazier, T. G., Thompson, C. M., and Dezzani, R. J.: A framework for the development of the SERV model: A Spatially Explicit Resilience-Vulnerability model, Applied Geography, 51, 158-172, 10.1016/j.apgeog.2014.04.004, 2014.

Please also note the supplement to this comment:
http://www.nat-hazards-earth-syst-sci-discuss.net/nhess-2016-408/nhess-2016-408-AC5-supplement.pdf
* * *
[Figure]

2017.

---

## Author Comment (AC6) · 23 May 2017

We are very grateful to the reviewer for the helpful comments on our manuscript. We have addressed all the comments made by the reviewer. To facilitate the review, we have modified the manuscript highlighting in yellow the changes carried out. We have taken advantage of this new opportunity to improve text and figures as the reviewer has requested. In this regard, the inclusion of exposure in the Integrated Social Vulnerability Index has been clarified, adding a new reference for this purpose. We have simplified the Figure 2 in order to improve its understanding. Moreover, we have explained better the concept of 'optimum number of clusters' at the results section, extending the information with regard to the BIC and the CAIC statistics. We have also modified the

text of the section 4.3 ('Policy implications'), giving some practical examples of specific mitigation measures that can be suggested for each cluster of urban areas. Finally, conclusions have been amended to express clearer how the methodology proposed here constitutes an improvement on the state of the art and the extent to which the results may be included in flood risk management plans, as both reviewers have recommended.

Reply to Anonymous Referee #2 comments and changes made

First of all, I very much enjoyed reading the manuscript. I have, however, a few comments to improve the manuscript. Please find them below.

Comment 1 - Abstract, line 16: 'it has not yet provided'. Please rephrase this a bit, the sentence is unclear.

Reply 1 - We have rephrased the sentence. As the reviewer pointed out, the sentence was badly phrased (page 1, lines 16-18).

Change 1 - "As far as we know, a methodological approach to construct the ISVI in urban areas of Castilla y León (northern central Spain, 94,223 km2, 2,478,376 inhabitants) prone to flash flooding has not yet been provided".

Comment 2 - some additional explanation is required on the inclusion of exposure in the social vulnerability index. In the traditional risk framework, exposure and vulnerability are two different components of the framework. As many researchers from the risk field read this journal, it should be specifically emphasized that including exposure is common practice in the social vulnerability field, even though this may contradict to the definition of risk and vulnerability which is more commonly used in the disaster risk community. This is important for the interpretation of the results.

Reply 2 - As the reviewer point out, exposure is commonly included in the social vulnerability analysis. Although exposure and vulnerability are two different components of risk, currently exposure is included in the social vulnerability assessments in order

to provide a holistic characterization of vulnerability, since it is not possible to talk about potential for loss (i.e. vulnerability) in the absence of exposure (Frazier et al., 2014). In the same way, resilience is also included in social vulnerability analysis, since the potential for loss also depends on the ability to absorb, cope with and recover from the effects of a disaster. Thus, a comprehensive social vulnerability assessment should include the social system's exposure and sensitivity to stress (exposure and sensitivity components of our Integrated Social Vulnerability Index, ISVI) as well as its capacity to absorb or cope with the effects of these stressors (resilience component of our ISVI) (Eakin and Luers, 2006). Therefore, we have explained that the inclusion of exposure in the social vulnerability analysis is a common practice (page 3, lines 7-9). We have also included a new reference in order to strength this idea.

Change 2 - "Although exposure and vulnerability are assumed to be two different concepts in the traditional risk framework, currently the inclusion of exposure as a factor to be considered is a common practice to assess social vulnerability".

Comment 3 - I have a few questions and a suggestion regarding Figure 2.

Change 3 - We have simplified the Figure 2 in order to make it clearer and easier to understand. Moreover, we have done a terminological change from 'municipalities' to 'urban areas', since municipalities is usually used to refer to administrative boundaries or local administration (i.e. the council). Thus, we have used the term 'urban areas' when we talk about the areas prone to flash flooding and 'municipalities' when we refer to the town halls.

Comment 3.1 - why is there an arrow going from Flash flood low probability municipalities to socioeconomic variables? Because the flash flood box is blue, it now seems like a hazard variable is added to the socioeconomic variables. This is, however, not the case (and should not be the case either).

Reply 3.1 - We wanted to show with this arrow that socio-economic variables had only been gathered to those urban areas that met the defined requirements, which were

then named as 'Flash flood low probability municipalities'. In order to facilitate the understanding of this part of the Figure 2, we have modified the color of this box from a blue plain to a gradient blue-beige color, indicating that the 'Flash flood low probability municipalities' box is the beginning of the second part of the figure (beige boxes).

Comment 3.2 - why are sensitivity and exposure 'clustered' and is resilience not in this cluster?

Reply 3.2 - All vulnerability components (exposure, sensitivity and resilience) were taken into account in the Latent Class Cluster Analysis (LCCA), using as input data the factor scores of the urban areas of interest. The minus sign which is placed on the arrow that goes from the box 'Factor scores' to 'Resilience' box indicates the sign of the component when the Integrated Social Vulnerability Index (ISVI) was calculated, and not that Resilience was not considered in the LCCA. Traditionally, factors that express sensitivity or exposure are considered as positive values in the ISVI (see the plus sign that is placed on the arrow that go from the box 'Factor scores' to 'Sensitivity' and 'Exposure' boxes); while factors that state resilience are considered as negative values, as has been done here.

Comment 3.3 - Perhaps add a third color that specifies the (final) results. This would make it more clear why some arrows exist in the framework (for instance the arrow from factor scores to clusters of municipalities).

Change 3.3 - Done. Thank you for the recommendation.

Comment 4 - Section 2.2.2: I do not fully understand the use of the Euclidian distance method. If I do understand it correctly, the sum of the differences between variable values is considered to be the distance? So distance is not spatial? I think it would be good to explain this a bit more clearly, as some parts of the paper are spatial (the clusters of municipalities for instance). This causes (at least for me) some confusion.

Reply 4 - We have changed the term 'distance' by 'similarity' in the text in order to clarify

that we were not talking about spatial distance (page 8, line 9). From a statistical point of view, distance measures are a type of similarity measure, so it is correct to use the term 'similarity'. We have used the Euclidean distance to evaluate how similar to each other the variables were, as is explained by Euclidean distance definition (page 8, lines 9-10). The greater the distance among variables are, the less similar the variables are. Hierarchical Segmentation Analysis (HSA) groups variables according to their similarity rather than the distance.

Change 4 - "Once the variables were standardized (Cutter et al., 2003), the squared Euclidean distance was used as a similarity measure, i.e., the square of the square root of the sum of the differences between variable values".

Comment 5 - Captions of Figure 4 and Figure 5 could be a bit longer. Figure + figure caption should be self-explanatory.

Reply 5 - We understand the comment about captions of Figure 4 and Figure 5, but we decided to shorten them in order to have a more balanced length of the text of this section.

Comment 6 - Figure 3 is perhaps not required, as it shows roughly the same as table 2? Perhaps move to appendix, as table 2 shows everything we would like to know (the variable clusters and the factor names)

Reply 6 - Figure 3 and Table 2 do not show the same information. Figure 3 corresponds to HSA output while Table 2 corresponds to the Factor Analysis (FA) output, which includes factor loadings that are necessary to construct the ISVI. HSA helps to overcome the Principal Components Analysis (PCA) sample size limitations, so we think that including the HSA output (i.e. the dendrogram) into the results section of this paper is an interesting approach.

Comment 7 - Section 3.2: I am a bit puzzled with the notion of 'optimum number of clusters'. What does an optimum amount of clusters mean? Ok the statistics say so,

but as a practitioner, what would it matter if you would have four clusters? How would this change the interpretation of the results?

Reply 7 - BIC and CAIC are statistics that enable to establish a number of clusters, which can be used in flood management. We used the BIC and the CAIC criteria in order to select the more parsimonious number of clusters (i.e. the number of clusters that provides as much information as possible taking into account the number of parameters to estimate). The more information is explained by the model, the greater the number of estimated parameters will be. The above is shown in Table 3 (Page 15). The minimum values of the BIC and the CAIC statistics are reached with a model of 3 clusters, and their values increase again in the estimations that consider four and five clusters. This means that the number of parameters to estimate by the model (see Table 3, 'Number of parameters' column) are too high in comparison to the obtained increase of explained information ('Log-likelihood(LL)' column). From a practical point of view, the above means that an increase in the number of clusters from 3 to 4 or 5 would split a fairly homogeneous cluster of urban areas into several subgroups which would not be very different from each other. Therefore, a greater level of disaggregation would not help to improve the implementation of different flood risk mitigation measures for each cluster of urban areas. Therefore, we have extended the information about the BIC and the CAIC criteria in the text (page 15, lines 2-3).

Change 7 - "The BIC and CAIC statistics were used in order to select the more parsimonious number of clusters (i.e. the number that provides as much information as possible taking into account the number of parameters to estimate)".

Comment 8 - Section 4.1: I would suggest to move parts of this to the method section. Most parts of this section are regarding the interpretation of the results. It is better to make this clear before the results section, instead of afterwards. A discussion after the results, weakens, in my opinion, the results.

Reply 8 - We have removed some parts of the text from section 4.1 instead of moving

the text because the text of the methodology section is already very long and it contains all the ideas that the authors considered important.

Comment 9 - Section 4.3: I suppose the clustering of municipalities is interesting from a policy making perspective. It would be good to link the clustering to this section. How can it improve policy making if we can identify similar municipalities?

Reply 9 - We have extended the text of the 'policy implications' section trying to emphasize the practical utility from a policy making perspective of the Latent Class Cluster Analysis (LCCA). For this purpose, we have included practical examples of specific mitigation measures that could be proposed for each cluster of urban areas identified by the LCCA (page 20, lines 16-26).

Change 9 - "The identification of social vulnerability patterns help to identify the most suitable mitigation measures for each cluster of urban areas identified by LCCA, in addition to prioritize the available resources. For instance, mitigation measures for those urban areas included in cluster 1 should be targeted towards improving the physical resilience (e.g. raising the first-floor elevation above ground level) and helping population financially with the implementation of mitigation measures (e.g. providing financial aid to those dwellings located at flood-prone areas). On the other hand, the population that live in those urban areas included in cluster 2 are highly dependent on external assistance due to high rates of ageing population, so emergency services should have adequately characterized the different evacuation routes (e.g. promoting the design of evacuation routes and the construction of shelters near those urban areas). Finally, mitigation measures for urban areas included in cluster 3 should be aimed at the collective facilities (e.g. practicing of flood emergency drills) and to encourage the implementation of individual mitigation strategies (e.g. through a financial incentive system, such as the repayment of part of the money spent at municipal taxes)".

Comment 10 - Please make the conclusions a bit more specific for this paper. What can we really learn from this paper, especially from a policy making perspective. What

does this paper add, besides being the first study on flash floods? A few lines on the conclusions for the study region (specific patterns identified) would be interesting as well.

Reply 10 - The conclusions have been reworded trying to make them more specific. They have been amended in order to express clearer how the methodology proposed here constitutes an improvement on the state of the art and the extent to which the results may be included in flood risk management plans and therefore improve flood risk management, which is the main objective of this social vulnerability analysis (page 21, lines 8-16).

Change 10 - A comprehensive characterization of social vulnerability is critical for an integrated FRM. The implementation of an HSA helps to overcome PCA sample size limitation, meaning an alternative methodology to the usually used to construct an ISVI in areas where available data is limited. The results show the high spatial heterogeneity of the social vulnerability within the study region and the high variability in the ISVI scores regarding the interactions between vulnerability components, which make an integrated analysis more important. The identification of vulnerability patterns through the LCCA gives the sources of vulnerability in each urban area, which simplifies the spatial heterogeneity analysis of the social vulnerability and enables to know what aspects need to be improved in order to decrease sensitivity and exposure and what aspects need to be reinforced to increase resilience. Thus, a better integration of the ISVI results into FRM plans and policies is made possible enabling to propose specific strategies of vulnerability reduction, increasing their efficiency.

CITED REFERENCES:

Eakin, H., and Luers, A. L.: Assessing the vulnerability of social-environmental systems, Annual Review of Environment and Resources, 31, 365-394, 10.1146/annurev.energy.30.050504.144352, 2006.

Frazier, T. G., Thompson, C. M., and Dezzani, R. J.: A framework for the development

of the SERV model: A Spatially Explicit Resilience-Vulnerability model, Applied Geography, 51, 158-172, 10.1016/j.apgeog.2014.04.004, 2014.

Please also note the supplement to this comment:
http://www.nat-hazards-earth-syst-sci-discuss.net/nhess-2016-408/nhess-2016-408-AC6-supplement.pdf

---

## Author Comment (AC7) · 23 May 2017

Dear Editor,

I am submitting a revised copy (in the track-change-mode) of our manuscript "Construction of an Integrated Social Vulnerability Index in urban areas prone to flash flooding" (doi:10.5194/nhess-2016-408) by Aroca-Jimenez et al. We are very grateful to the reviewers for their helpful comments on our manuscript. We have addressed all the comments made by the reviewers. To facilitate the review, we have modified the manuscript highlighting in yellow the changes carried out. We have taken advantage of this new opportunity to improve text and figures as the reviewers and the editor have requested. In this regard, the concept of both vulnerability and all its components (i.e.

[Figure]

sensitivity, exposure and resilience) have been clarified. In addition, the inclusion of exposure in the Integrated Social Vulnerability Index has been clarified, adding a new reference for this purpose. As the reviewers recommended, we have simplified the Figure 2 in order to improve its understanding. We have also created a new subsection under the section 2 (i.e. "2.2.2 Database generation"). Moreover, we have explained better the concept of 'optimum number of clusters' at the results section, extending the information with regard to the BIC and the CAIC statistics. We have modified Figure 3 by adding the description of the variables in order to increase readers' friendliness as the reviewer suggested. To facilitate understanding of the results, we have added a new column to Table 2, indicating the vulnerability component to which each vulnerability factor belongs. Furthermore, we have modified the text of the section 4.3 ('Policy implications'), giving some practical examples of specific mitigation measures that can be suggested for each cluster of urban areas. Finally, conclusions have been amended to express clearer how the methodology proposed here constitutes an improvement on the state of the art and the extent to which the results may be included in flood risk management plans, as the reviewers have recommended.

We thank you for the opportunity to resubmit our manuscript to the journal Natural Hazards and Earth System Sciences and hope that it is now suitable for publication. We look forward to hearing from you at your earliest convenience.

Best regards,

Estefania.

Please also note the supplement to this comment:
http://www.nat-hazards-earth-syst-sci-discuss.net/nhess-2016-408/nhess-2016-408-AC7-supplement.pdf

**Supplement:**

[revised manuscript text omitted]

---

## Author Comment (AC8) · 23 May 2017

Dear Editor,

I am submitting the new version of our manuscript "Construction of an Integrated Social Vulnerability Index in urban areas prone to flash flooding" (doi:10.5194/nhess-2016-408) by Aroca-Jimenez et al. after including the improvements suggested by the reviewers.

We thank you for the opportunity to resubmit our manuscript to the journal Natural Hazards and Earth System Sciences and hope that it is now suitable for publication. We look forward to hearing from you at your earliest convenience.

Best regards,

Estefania.

Please also note the supplement to this comment:
http://www.nat-hazards-earth-syst-sci-discuss.net/nhess-2016-408/nhess-2016-408-AC8-supplement.pdf

**Supplement:**

[revised manuscript text omitted]

---

## Author Response (AR2)

**Reply to Anonymous Referee and Editor comments and changes made**

We are very grateful to the reviewer and the editor for the helpful comments on our manuscript. We have improved the manuscript including some of the comments we already did for the previous step of the peer-review process, as the reviewer and the editor have recommended.

Most of my concerns about the paper were addressed in the supplement files with the comments for the reviewers but some of the explanations are not reflected on the revised version of the manuscript. In this line, it can be now accepted for publication with minor revisions. In these revisions, the authors should try to improve the manuscript based on the comments that they did for the reviewers and to give some better explanations on different parts of the manuscript.

Details of the changes made are found below:

- We have given a clear framework of the vulnerability concept used in this study (page 2, lines 21-28).

**Currently, the most used approach for analyzing vulnerability is a hybrid approach between risk-hazard approaches, which considers that vulnerability depends on the biophysical risk factors and the potential loss of a particular exposed population (e.g. the Hazards-of-place model of vulnerability (Cutter, 1996); and political economy/ecology approaches, which emphasize the political, cultural and socioeconomic factors that explain the differential exposure, impacts, and capacities to recover from an impact (e.g. the Pressure and Release model (Blaikie et al., 1994). Taking into account the key parameters for the vulnerability research that highlight the above-mentioned approaches, vulnerability depends on the social system's exposure and sensitivity to stress (i.e., any characteristics that increase vulnerability) as well as its capacity to absorb or cope with the effects of these stressors (i.e., resilience (Adger, 2006; Eakin and Luers, 2006;** Birkmann, 2013; Thieken et al., 2014).

- We have explained better what the authors understand as integrated vulnerability (page 3, lines 10-11).

Less attention has been paid to integrated analysis of vulnerability components (**i.e., using the hybrid approach above mentioned**), which considers [...].

- We have added a comment about the inclusion of exposure in the social vulnerability index (page 3, lines 18-20).

[...], currently the inclusion of exposure as a component to be considered is a common practice to assess social vulnerability **in order to provide a holistic characterization of vulnerability** (Turner et al., 2003; Adger et al., 2004), **and it is not possible to talk about potential for loss (i.e., vulnerability) in the absence of exposure (Frazier et al., 2014).**

- We have clarified why we used a low probability scenario (i.e., flood hazard zones with low or exceptional probability) (page 6, lines 11-12).

[...], we located the urban environments defined by Basin Water Authority as Areas with Potential Significant Flood Risk (APSFRs) (Caballero et al., 2011) and **the flood hazard zones with low or exceptional probability (i.e., 500-year flood), using the low probability scenario because it is the most comprehensive representation of urban areas that could be affected by flash floods at regional scale.**

- We have added additional text to clarify that the Hierarchical Segmentation Analysis (HSA) does not use spatial distance (page 8, lines 6-7).

**The greater the distance among variables, the less similar the variables are.**

- We have explained better the idea behind the equation's modification from the original one presented by Frazier et al. (2014) (page 9, line 18).

[...], although the tolerance statistic was used here as a weighting method **instead of the amount of explained variance**.

- We have explained better the practical implications of considering 3 clusters as the optimum number of clusters (page 20, lines 3-8).

**BIC and CAIC criteria enable to establish the optimum number of clusters (i.e., 3 clusters in this case). From a practical point of view, the above means that an increase in the number of clusters from 3 to 4 or 5 would split a fairly homogeneous cluster of urban areas into several subgroups which would not be very different from each other. Therefore, a greater level of disaggregation would not help to improve the implementation of different flood risk mitigation measures for each cluster of urban areas.**

- We have rewritten the section 'Author contribution' in order to express better the contribution of each author (page 21, lines 25-28).

[revised manuscript text omitted]

---

## Author Response (AR4)

**Reply to Editor comments and changes made**

We are very grateful to the editor for the helpful comments on our manuscript. We have improved the manuscript including the changes that the editor has recommended. We have maintained the changes introduced at the previous version of the manuscript because some of the comments have been improved (changes highlighted in green in the manuscript). On the other hand, changes related to language have been highlighted in yellow. The changes made are explained in detail below:

1) Abstract - Language editing by a native speaker is needed. Moreover the abstract miss a red line and have to be restructured.

The whole text has been revised by a native speaker. Changes related to the language have been highlighted in yellow. Thank you very much for your suggestion. We hope that the changes made have improved the understanding of the manuscript.

2) This sentence could be interpreted that risk only depends on different hazard types and this contradicts with your approach and the concept of risk. Please revise this sentence.

The sentence has been modified in order to avoid confusion (page 1, line 10).

Among the natural hazards, flash flooding is the leading cause of weather-related deaths.

3) This relation is not clear.

After the changes made in the previous sentence, we think that the relation is expressed in a clearer way (page 1, lines 10-11).

4) Please formulate your sentence more precise and not as a fact. Everything depends on the applied definition.

We have modified the sentence (page 1, lines 11-13). Thank you for the suggestion.

In this regard, integrated social vulnerability (ISV) can incorporate the spatial distribution, contribution [...]

5) Wording: characterized characteristics.

We have modified the phrase (page 1, lines 13-15). Thank you for the recommendation.

ISV is defined by the demographic and socioeconomic characteristics that condition [...]

6) I recommend to reduce the number of citations.

We have reduced the number of citations as the editor has recommended (page 1, lines 27-28).

7) This is not a good way to bring up a definition which is very important for the study. Make one sentence of this part.

We have moved the definition of the vulnerability assessment (page 2, lines 18-20).

Vulnerability assessment can be defined as analysis of the characteristics of a person or group and their situation that influence their capacity to anticipate, cope with, resist and recover from the impact of a natural hazard (Birkmann, 2013).

8) Do you mean steep slopes?

We have modified this word (page 4, line 11). Thank you for the correction.

Steep slopes, which limit [...]

9) I do not understand this sentence. In which context are the differences? population density?

There are large differences between rural and urban areas in the region of Castilla y León, despite they are the same administrative unit (i.e., municipalities). These differences are related, among other things, to population density, access to medical services or municipal budget. However, we have rewritten the phrase in order to avoid confusion (page 4, line 15).

The region is divided into 2,248 urban areas.

10) Sources of the maps are missing. Check quality of figures regarding the needed resolution (Figure 1).

We have added the source of the maps as the editor has recommended. Moreover, we have checked quality of all figures reading the guide for authors.

Figure 1 is submitted in *.png* format, it has a resolution of 300 dpi, as the 'Manuscript preparation guidelines for authors' indicates.

11) Rewrite this sentence.

Done. Thank you very much for the recommendation (page 5, lines 9-11).

It was not possible to perform a PCA to define the ISVI, as  is usual in the literature (Fekete, 2010; Frazier et al., 2014;Hummell et al., 2016), since the number of variables initially considered (71) outnumbered the urban areas of interest (39) (Sarstedt and Mooi, 2014).

12) Provided the year of all these datasets.

Done. Thank you very much for the suggestion (page 6, lines 10, and page 7, lines 1 and 2).

It was provided by the Spanish National Geographic Institute (IGN; **layer generated in 2013**) and was used as input data for the Geospatial Hydrologic Modelling extension (HEC-GeoHMS 10.0) (USACE, 2013) from which the longitudinal slopes of the river were calculated. Secondly, we examined urban environments

defined by the Basin Water Authority as Areas with Potential Significant Flood Risk (APSFRs; **layer generated in 2015**) (Caballero et al., 2011) and flood hazard zones with low or exceptional probability (i.e., 500-year flood; **layer generated in 2016**), taking into account the river reaches selected in the previous step.

13) Provide 2 or 3 examples of references.

Done. Thank you very much for the suggestion (page 7, line 11).

Based on existing literature (Cutter et al., 2003; Frazier et al., 2014; Hummell et al., 2016), a set of 71 variables was initially characterized for each of the 39 urban areas identified above.

14) Which type of questionnaires did you use (structured interview?).

We did not use a specific type of questionnaire, but we telephoned directly the councils or public organizations and we asked for the data we wanted to know (page 7, lines 14-15).

15) How did you to that, perhaps add this information with a table in the appendix

We have added this information into the 'Data source' column of the Table 1 (Table 1).

16) How did you generate information on collective vulnerability? How do you define collective vulnerability?

The term 'collective vulnerability' encompasses the vulnerability aspects that are related to the community as a potentially sensitive unit (Adger, 1999). Variables contained in 'collective vulnerability' category were 'Potential intersections between evacuation routes and rivers' and 'Areas suited to population evacuation', and they were obtained through GIS analysis (see Table 1). We have added the definition of 'collective vulnerability' (page 7, lines 17-18).

17) Please define this more precise (Table 1).

We have detailed in 'Data source' column of Table 1 how we got all the variables. We hope this information is expressed clearer now in the manuscript. Thank you for the recommendation.

18) Please explain in the text the possible effects of combing data of different years, e.g. amount of buildings and information to households.

It is very difficult not to include data from different years, since each public agency has its own mechanisms for updating data. This was partly corrected by using relative values of variables instead of absolute values. We have included data from the year 2009 to 2015, so that we consider that the social structure has not changed in this very short period of time. We have added this explanation in the discussion section (page 19, line 13 and page 20, lines 1-2)

It is also worth mentioning that it is very difficult not to include data from different years in this type of analysis related to flash floods, since different databases are usually consulted and each public agency has its own mechanisms for updating data.

19) Rephrase this sentence

We have rephrased the sentence. Thank you for the suggestion (page 9, lines 12-14).

The number of groups was determined by taking into account both the distance at which groups were differentiated into the graphical output of the HSA (i.e., the dendrogram) and the consistency and homogeneity of the numbers of variables contained in them.

20) What is a collective building?

We have named 'collective buildings' to all those large facilities, such as schools, hospitals, retirement homes, etc. We have changed the term used in the text in order to avoid confusion (page 11, line 7).

The first group contained variables mainly related to large facilities.

21) Indicate these in the figure.

The fifth group showed bilateral correlations (i.e., pairs of variables), therefore all variables displayed significant correlations with at least one variable. We have decided not to modify Figure 3 because we should have had to highlight all variables of this group, which did not add valuable information (Figure 3). We have modified the text in order to avoid confusion (page 11, line 11).

22) Increase and check quality of figure. Provide sources of maps (Figure 4).

We have added the sources of the map. Moreover, we have checked the quality of the figure. All maps that compose Figure 4 are submitted in *.png* format and they have a resolution of 300 dpi. Moreover, Figure 4 has been improved increasing the font size of map legends.

23) Whole discussion section has to be more related to your results and compared to other studies. You can discuss here more in depth the difference of the regions or what are the changes to a traditional FRM in these different urban areas. Be more specific.

The followed strategy for writing discussion section of the manuscript was to match the obtained results with the socio-economic and demographic characteristics of the study area. We selected this strategy because there is no scientific consensus regarding the methodology needed for constructing social vulnerability indexes, so there are almost as many types of indexes as published works. This makes the comparison between different social vulnerability indexes difficult. Thus, the discussion of the paper not only explains the results of this study, but also tries to match the obtained results with specific flood risk mitigation measures that could be implemented by competent public agencies according to the characteristics of the study area.

24) Please provide more insights of your study in this context (limitation, challenges, uncertainty ...) and compare this with other studies and approaches. This sub-section is very similar to other parts where you highlighted the gaps (section 4.1).

We have tried to explain the main limitations and challenges faced in the constructions of our ISVI compared to social vulnerability indexes designed in the natural disaster field. Limitation and challenges explained in the text are listed below:

**Limitation 1. Information available in mountainous urban areas is usually limited (page 19, lines 7-12).** Flash floods usually affect small mountainous urban areas where the information available is limited, either because it is not available in public databases or because it is not generated on this work scale. This type of urban areas are usually managed by small councils, which do not usually have sufficient resources to generate the required information. This imposes limits on any assessment related to flash floods. However, this constraint does not usually apply to studies on fluvial floods since, in terms of population, these frequently affect significant urban areas, which generally means greater availability of data and a larger number of event records.

**Limitation 2. The lack of information can condition the selected work scale (page 20, lines 1-5).** The lack of information may condition the selected work scale (i.e., it is not possible to work at a certain work scale if there is no information about the main variables needed for constructing the ISVI), which could result in homogeneous vulnerability reduction measures being put in place in areas where the spatial variability of vulnerability is high. This would reduce their effectiveness and might not guarantee a uniform reduction in vulnerability. In this study, the selected work scale was the urban area, as this entity tends to be small and homogeneous in the region of Castilla y León, especially if we are talking about mountainous urban areas.

**Limitation 3 and challenge 1. Sensitivity and resilience are considered as static components (page 20, lines 6-10).** The results give a snapshot of vulnerability, however sensitivity and resilience can vary over time and space. The identification of spatial patterns in this work represents a step forward towards improving FRM on a regional scale. Regarding temporal variability, we suggest periodic monitoring of identified variables as an explanation of social vulnerability to flash floods. Periodic recalculations would allow urban areas to keep informed about the behaviour of SVI values over time (here we are talking about a challenge).

**Limitation 4. ISVI values are not absolute (page 20, lines 12-14).** ISVI can be used qualitatively to determine whether one urban area is more vulnerable than the others and, if so, to what extent. Methodology of this work can be extrapolated to other urban areas prone to flash flooding, but it is very difficult to extrapolate the results because they characterize the vulnerability of a certain study area under certain socio-demography and socio-economy characteristics.

**Limitation 5. The PCA sample size limitation (page 20, lines 15-20).** Conducting a preliminary HSA helps overcome the limitations of the PCA sample size. Most published works either do not discuss this aspect or tackle it by adding the variables directly. However, the HSA enables the vulnerability variables to be divided into groups, which already allows the implementation of a PCA in each group for constructing the ISVI. Nevertheless, HSA did not provide information on the relative significance of variables within each group, making it necessary to subsequently perform a PCA.

**Limitation 6 and challenge 2. Different views regarding the weighting method and the spatial variability of the vulnerability factors (page 18, lines 20-25).** It seems reasonable to suppose that not all factors have the same importance in the construction of the ISVI, especially when there may be variations in the number of variables forming each factor and their explained variance. It is even possible that there is a

spatial variation in each factor's importance. This can be solved by carrying out geographically weighted principal component analysis (GWPCA), which would need more detailed information.

25) Rephrase - is very unclear

We have rephrased this part (page 19, lines 11-13). Thank you for the recommendation.

However, this constraint does not usually apply to studies on fluvial floods since, in terms of population, these frequently affect significant urban areas, which generally means greater availability of data and a larger number of event records.

26) But this is then also a snapshot

They were effectively snapshots, but it is possible to perform temporal analysis with a set of snapshots and to obtain more reliable results. This also happen with other variables, such as river flows. It is possible to have one single data of river flow or a dataset of river flows, which enable to conduct more comprehensive analysis (page 20, lines 11-13).

27) Rephrase

Done (page 20, line 15). Thank you for the recommendation.

As regards calculation of the ISVI, it is crucial that ISVI values should not be considered as absolute.

28) Rephrase

Done (page 20, lines 17-18). Thank you for the suggestion.

In the methodology proposed here, conducting a preliminary HSA helps overcome the limitations of the PCA sample size.

29) This is unclear. Do you mean that flooding have an higher impact on these type of buildings?

Given the vulnerability factors obtained, we can know that exposure component is partly characterized by dwellings with basement (see 'exposure in the urban built-up environment' factor of Table 2) and households with 1 storey above ground level (see 'constructive exposure' factor of Table 2). Flooding have a higher impact on these type of buildings in comparison with households with two or more storeys above ground level in two ways: i) people that live in this type of dwellings have not an upper floor where they can shelter during the flood; and ii) content damages can be higher in this type of buildings since most of household items can be reached by the flood (page 21, line 8).

30) How this is related to your results? Can you proof the interpretation of other studies in your study? Same questions for the paragraph above (section 4.2).

We explain the obtained results in this section, so it is fully related to our results. We have added the name of the vulnerability factors in the different paragraphs in order to facilitate the understanding of the section and to link results (Table 2 and Figure 4) with discussion clearer.

Furthermore, bibliographic references included in this section not only support the expressed ideas, but also in a way compare the variables obtained in other studies with the ones we have obtained in our work.

31) That is on very general level. Be more specific related to your study. Are the result used now by decision makers? (section 4.3)

There is a general introduction in this section talking about possible policy implications of this work. However, we already added examples of possible flood risk mitigation measures of each cluster of urban areas identified by Latent Class Cluster Analysis (LCCA) in order to be more specific and show clearer the practical implications of our results. Examples that were already introduced are highlighted below:

**Cluster 1.** [...] mitigation measures for those urban areas included in cluster 1 should be targeted towards improving physical resilience (e.g. raising the first-floor elevation above ground level) and giving the population financial help to put mitigation measures in place (e.g. providing financial aid for dwellings located in flood-prone areas).

**Cluster 2.** [...] people living in the urban areas included in cluster 2 are highly dependent on external assistance due to the high rates of ageing population. Therefore, different evacuation routes should be designed and clearly defined by the emergency services, and shelters constructed near these urban areas.

**Cluster 3.** [...] mitigation measures for urban areas included in cluster 3 should be aimed at collective facilities (e.g. carrying out flood emergency drills) and should encourage the implementation of individual mitigation strategies (e.g. through a financial incentive system, such as repayment of part of the money spent on municipal taxes).

32) Rephrase whole sentence - relation does not match. How can population sensitivity be damaged by a flash flood?

Sensitivity is the term used in bibliography to refer to the level to which people and assets can be damaged by a flash flood. The main objective of the ISVI was to identify those socio-economic and demographic characteristics that influence the population capacity to anticipate, cope with, resist and recover from the impact of a natural hazard. Therefore, the ISVI enables to know which characteristics make that a certain population is more damaged than other for the same flash flood event (page 22, lines 21-23).

33) Can you proof this for your study?

Castilla y León is a very large region with a low population density. In addition, mountainous urban areas prone to flash flooding usually are very small areas with few structures and facilities, and limited economic activities. Therefore, resources in this type of areas are limited, especially economic resources, as well as the access to specific public services (page 23, lines 9-10).

34) Rephrase whole sections. Overall, please try to formulate shorter and more clear sentences.

We have rewritten the whole section (page 23, lines 21-29). Thank you for the recommendation.

A comprehensive characterization of social vulnerability is critical for an integrated FRM. The implementation of an HSA helps to overcome PCA sample size limitation. This means using an alternative methodology to the one usually used to construct an ISVI in areas where available data is limited. The results show the high spatial heterogeneity of social vulnerability within the study region and the high variability in ISVI scores regarding interactions between vulnerability components, which give integrated analysis greater importance. The identification of vulnerability patterns through LCCA gives the sources of vulnerability in each urban area. This simplifies the spatial heterogeneity analysis of social vulnerability and indicates which aspects need to be improved to decrease sensitivity and exposure and which aspects need to be reinforced to increase resilience. This allows the ISVI results to be more effectively integrated into FRM plans and policies, which in turn enables specific strategies of vulnerability reduction to be proposed, thereby increasing their efficiency.

**CITED BIBLIOGRAPHY**

Adger, W. N.: Social vulnerability to climate change and extremes in coastal Vietnam, World Development, 27, 249-269, 1999.

[revised manuscript text omitted]